# The Role of Regional Feedbacks in Glacial Inception on Baffin Island: The Interaction of Ice Flow and Meteorology

Leah Birch[1], Timothy Cronin[2], and Eli Tziperman[1]

[1]School of Engineering and applied Sciences and Department of Earth and Planetary Sciences, Harvard University, Cambridge, MA, USA

[2]Department of Earth, Atmosphere and Planetary Sciences, MIT, Cambridge, MA, USA

*Correspondence to:* Leah Birch (lbirch@seas.harvard.edu)

**Abstract.** Over the past 0.8 million years, 100 kyr ice ages have dominated Earth's climate with geological evidence suggesting the last glacial inception began in the mountains of Baffin Island. Currently, state-of-the-art global climate models (GCMs) have difficulty simulating glacial inception, possibly due in part to their coarse horizontal resolution and the neglect of ice flow dynamics in some models. We attempt to address the role of regional feedbacks in the initial inception problem on Baffin Island by asynchronously coupling the Weather Research and Forecast model (WRF), configured as a high resolution inner domain over Baffin and an outer domain incorporating much of North America, to an ice flow model using the shallow ice approximation. The mass balance is calculated from WRF simulations, and used to drive the ice model, which updates the ice extent and elevation, that then serve as inputs to the next WRF run. We drive the regional WRF configuration using atmospheric boundary conditions from 1986 that correspond to a relatively cold summer, and with 115 kya insolation. Initially, ice accumulates on mountain glaciers, driving downslope ice flow which expands the size of the ice caps. However, continued iterations of the atmosphere and ice models reveal a stagnation of the ice sheet on Baffin Island, driven by melting due to warmer temperatures at the margins of the ice caps. This warming is caused by changes in the regional circulation that are forced by elevation changes due to the ice growth. A stabilizing feedback between ice elevation and atmospheric circulation thus prevents full inception from occurring.

## 1  Introduction

For the past million years, Earth's climate has repeatedly fallen into ice ages with a periodicity of $\sim 100,000$ years (Petit et al., 1999). How glacial inceptions occur is still a largely unresolved problem in climate science (Rind et al., 1989; Otieno and Bromwich, 2009). The leading hypothesis for the cause of glacial inception is that orbital variations (Milankovitch, 1941) cause cooler summers leading to less melting, although previous studies found that additional amplifying feedbacks beyond the ice albedo feedback are necessary to explain inception. Possible amplifiers of orbital forcing may be divided into large-scale non-local feedbacks, including ocean temperature and planetary-scale circulation changes, and local feedbacks associated with greater height and extent of mountain glaciers (Lee and North, 1995; Oerlemans, 2002; Abe-Ouchi et al., 2007), regional circulation changes, clouds, and more. In this study, we seek to understand specifically the role of local ice-climate feedbacks in glacial inception using a regional climate model asynchronously coupled to a simple ice sheet model. Our regional model is

centered over Baffin Island – the likely location of the last glacial inception (Ives, 1962; Ives et al., 1975; Andrews and Barry, 1978; Williams, 1979; Clark et al., 1993).

Glacial inception of the Laurentide ice sheet is thought to have progressed fairly rapidly from the Baffin Island in eastern Canada (Clark et al., 1993), with the associated drop in sea level estimated to be as large as 50 m in 10kyrs (Waelbroeck et al., 2002; Ganopolski et al., 2010). The Laurentide ice sheet has been identified as the primary initial cause of the drop in sea level (Kleman et al., 2013) and some ice modeling studies have shown rapid inception within a few thousand years (Andrews and Mahaffy, 1976; Marshall and Clarke, 1999). However, this rapid growth required extreme precipitation increases or excessive climate cooling every thousand years, without any explicit explanation. A more modest sea level drop 10-20 m in 10 kyrs has also been posited (Lambeck and Chappell, 2001; Creveling et al., 2017), which indicates a slower ice sheet growth rate and possibly ice growth relegated to Arctic Canadian Islands (Kleman et al., 2002).

Global climate models (GCMs) have been used to simulate glacial inception, focusing on large-scale feedbacks. They often find that snow does not accumulate in unglaciated areas, or that glaciers do not spread into unglaciated areas. One reason GCMs have difficulty simulating the glacial inception is their coarse horizontal resolution, which lowers the topographic features dramatically (Wang and Mysak, 2002; Vettoretti and Peltier, 2003). Snow then does not accumulate because the region is too warm due to artificially low elevations (Rind et al., 1989). Though topography is important (Oerlemans, 2002; Marshall and Clarke, 1999; Birch et al., 2017), other aspects of the climate system may also be misrepresented in GCMs, like ocean feedbacks (Khodri et al., 2001), dust (Calov et al., 2005; Farrell and Abbot, 2012), sea ice (Gildor and Tziperman, 2001; Li et al., 2005), vegetation (Wang et al., 2005; Goñi et al., 2005), and meteorology (Bromwich et al., 2002a; Jackson and Broccoli, 2003; Birch et al., 2017). Increasing the model resolution has lead to moderate improvements in the simulation of glacial inception (Jochum et al., 2012; Vavrus et al., 2011; Otieno et al., 2012), through the occurrence of areas of net snow accumulation, which can progressively expand as the climate cools. Given that GCMs had such difficulties in simulating glacial inception, our study focuses on the importance of regional feedbacks over the Baffin island.

While Milankovitch forcing explains some summer cooling, the large scale circulation is an important driver of heat and moisture fluxes to inception areas. In the current climatology, large scale cyclonic circulation over Baffin Bay is associated with cool summers (Bromwich et al., 2002a; Gardner and Sharp, 2007), which may be all that is necessary for inception. However, Jackson and Broccoli (2003) and Khodri et al. (2001) also noted increased storms and poleward moisture fluxes during inception. Otieno and Bromwich (2009) showed that in addition to favorable circulation and temperature, snow could accumulate in new areas only after artificial large-scale cooling was prescribed. The question remains: how does this cooling occur? One possible answer is the interaction of ice sheets and the atmosphere.

As the ice sheet grows, it influences the atmosphere due to albedo and surface elevation changes. The regional circulation can also be diverted by the changing topography. Anti-cyclones have been noted to occur over mountains (Smith, 1979), but ultimately, the effects of mountains on the regional climate and large-scale circulation depends on the mean overlying flow and mountain size (Chen and Lin, 2005; Cook and Held, 1988). Looking at large-scale ice growth in the context of glacial inception, Roe and Lindzen (2001a) argued using a simple model that slight changes in topography can influence the stationary waves, significantly changing the circulation and leading to colder temperatures. The topography and geographic location may

be critical to the glacial inception process (Jackson, 2000; Abe-Ouchi et al., 2013), but previous studies did not consider the effect of ice-growth induced regional topography changes on Baffin Island.

To capture the interaction of ice and the atmosphere, climate models of intermediate complexity have been combined with ice sheet models. This type of coupled ice-climate model allows thousands of years of ice growth due to changes in insolation and $CO_2$ to be simulated. Often, however, these models glaciate Alaska and Western Canada very rapidly (Kageyama et al., 2004; Bonelli et al., 2009; Ganopolski et al., 2010), which is not consistent with the geological record showing inception occurred first over the Baffin Island in eastern Canada (Williams, 1979; Clark et al., 1993). Ice sheet models and GCM studies (Dong and Valdes, 1995; Meissner et al., 2003) have a similar problem of not agreeing with the geological record. Combining GCMs and ice sheet modeling may allow for progress in understanding ice-atmosphere interactions (Herrington and Poulsen, 2011; Gregory et al., 2012; Löfverström et al., 2014), but the smoothed topography and other biases in these coarse resolution climate models often need to be artificially corrected to obtain an accurate regional climate. Furthermore, mountain ice caps of the Canadian Arctic Archipelago in the present climate are too small to be modeled by large-scale ice models, hence our use of a higher-resolution regional ice model.

The interaction of the ice and the atmosphere is a critical component of glacial inception, often left out of climate models. For instance, Birch et al. (2017) focused on a single ice cap on Baffin Island using a high-resolution cloud-permitting regional climate model without an ice flow model. That study found that ice would accumulate on the mountain glaciers of Baffin Island. The 4 km resolution of that study captured both the ablation and accumulation zones of mountain glaciers. By integrating the mass balance from the top of the ice cap down slope, Birch et al. (2017) implicitly represented the effects of ice flow. As a result, the Penny Ice Cap was deduced to grow outside of its present day bounds due to decreased insolation and a sequence of cold summers and wet winters. Consistent with previous GCM studies, no snow accumulated in unglaciated areas.

Thus, the purpose of the current paper is to study the role of regional feedbacks in the inception over the major ice caps of Baffin Island, including the role of ice flow. By prescribing boundary conditions from the modern climate, we do not consider any role of non-local ice sheet growth in modifying the planetary wave pattern, which could aid or hinder inception on Baffin Island. This is consistent both with other studies (Bonelli et al., 2009; Ganopolski et al., 2010) and with our intention to study the effect of regional feedbacks. Similarly, we neglect the role of SST changes, as there is debate over warm (Stokes, 1955; Ruddiman et al., 1980; Cortijo et al., 1994; Gildor and Tziperman, 2000, 2001) or cold oceans during inception (Khodri et al., 2001; Lehner et al., 2013). We choose to keep modern SSTs and identify whether inception may be described as a local process over Baffin Island, driven by Milankovitch forcing. Regional ice-albedo and height-mass balance feedbacks are expected to dominate at least some of the planetary scale feedbacks at the very beginning of the inception process. We drive the model with boundary conditions corresponding to wet and cool summers from reanalysis, ideal for inception but not extremely uncommon during present-day climate. Such conditions may be the result of SST forcing that is not explicitly simulated, so in that sense we do include some SST effects. Our regional model configuration could also be driven by the boundary conditions from a GCM simulation of glacial inception, which would incorporate some planetary-scale feedbacks. However, given the inability of such models to simulate the inception and often the lack of substantial circulation changes over Baffin Island (Otieno et al., 2012), we choose not to take that route and focus solely on regional feedbacks.

We investigate the feedbacks between a growing ice sheet and the atmosphere at higher spatial resolution than can be simulated with GCMs, using an shallow ice approximation ice flow model asynchronously coupled with the Weather Research and Forecasting model (WRF, Skamarock et al., 2008). We find that the change to ice sheet topography induces an anticyclonic atmospheric response over Baffin Island. This anticyclone causes a flow of warm air over Baffin Island and Hudson Bay area, increasing ablation, stagnating further ice sheet growth, and thus indicating a negative feedback on inception at the regional scale.

The atmospheric and ice models used for the asynchronous coupling experiment are presented in Section 2. We present the results in Section 3, beginning with the mass balances and resulting ice thickness of our atmosphere-ice iterations. We then investigate the regional climate and local circulation impacts of expanding ice cover and increasing ice elevation. Sensitivity simulations exploring the importance of topography and insolation are also presented. Finally, the sensitivity of ice flow to ice viscosity and mass balance are examined. We conclude with a discussion in Section 4, including the limitations of this study and its implications.

## 2 Methods

Our goal in this study is to better understand the interaction between changing ice cover and the regional climate of Baffin Island and the feedback on the regional circulation over Canada. We therefore use a nested configuration of the Weather Research and Forecasting Model (WRF Skamarock et al., 2008) with the outer domain covering all of North America with a 100 km resolution and the inner domain covering Baffin Island with a 20 km resolution, encompassing both the Penny and Barnes Ice Caps. To better understand the growth of these mountain glaciers during inception, we must simulate ice flow in the inner 20 km resolution domain, but WRF does not contain an ice flow model, which means the ice expansion must be calculated offline. Based off of Oerlemans (1981), we developed a simple ice flow model that reduces the 3D Stokes flow to two dimensions using the shallow ice approximation. The basal sliding velocity is assumed to be 0. This approximation results in a nonlinear diffusion equation that represents ice flow by evolving ice thickness H in time based on horizontal gradients of surface elevation H*, along with surface mass balance G:

$$\frac{\partial H}{\partial t} = \nabla \cdot D \nabla H^* + G \tag{1}$$

In Equation 1, the yearly surface mass balance $G$ is calculated from WRF simulations and assumes that the density of ice is $900 \, \text{kg m}^{-3}$. The surface elevation $H^*$ and ice thickness $H$ are linked by $H^* = B + H$, where $B$ is the basal topography. The nonlinear shear-thinning of ice flow is represented by a diffusivity, $D$, which depends on both the ice thickness and surface elevation gradients:

$$D = AH^{m+2}\left[\left(\frac{\partial H^*}{\partial x}\right)^2 + \left(\frac{\partial H^*}{\partial y}\right)^2\right]^{(m-1)/2} \tag{2}$$

The parameter $m$ for all model simulations is set to 3, which is a typical value for realistic ice simulations (Cuffey and Paterson, 2010). The coefficient $A$ in Equation 2 is the creep parameter that influences the viscosity of the ice is based on the temperature of the ice, i.e. the colder ice flows more slowly, while warmer ice more quickly. We use a creep parameter $A = 3.5 \times 10^{-25}$ Pa$^{-3}$s$^{-1}$ from Cuffey and Paterson (2010). More details on this choice of creep parameter can be found in Appendix A, which also contains the details of the numerical scheme used to integrate this ice model in time.

The ice model requires the specification of initial surface elevation ($H^*$), initial ice thickness ($H$), basal topography ($B$) that is time-independent, and mass balance ($G$) as an input for each time step. The ice thickness ($H$) is adapted from the present day ice cover on Baffin Island from NASA Operation IceBridge Mission (Kurtz and Farrell, 2011), while $H^*$ and $G$ are taken from our WRF simulations. The grid size and time step are also user specified and chosen to be 20 km (same as WRF inner domain) and 1-year, respectively. With these atmospheric and ice models, we perform iterations, running WRF, using the mass balance calculated by WRF to force the ice model, updating the ice thickness and extent calculated by the ice model, and then running WRF again with the newly calculated ice cover as an input. This amounts to an asynchronous coupling of the two models, where a WRF run followed by the ice model is considered an iteration.

WRF (Skamarock et al., 2008), version 3.7, is a compressible, non-hydrostatic model, which has been shown to provide a reasonable simulation of mountainous and Arctic regions (Cassano et al., 2011; Kilpeläinen et al., 2011). Reanalysis data from the European Centre for Medium-Range Weather Forecasts (ECMWF, Dee et al., 2011) are used for setting the meteorological boundary and initial conditions, including sea surface temperatures and sea ice extent. Cassano et al. (2011) also noted that the Interim reanalysis (ERA-Interim) is a suitable choice of boundary conditions for WRF in the Artic.

We use the WRF Single Moment 6 Class microphysics scheme, which allows for ice, snow and graupel processes (Hong and Lim, 2006). NOAH-MP (Niu et al., 2011) is used for the land surface model. It includes 4 layers of snow and allows for refreezing, which is an important process on the Penny Ice Cap according to historical trends (Zdanowicz et al., 2012). NOAH-MP caps the snow water content at 2000 kg m$^{-2}$ by default, but we remove this limit. The radiation scheme is the Rapid Radiative Transfer Model for GCM applications (RRTMG, Iacono et al., 2008), which has been found to have minimal bias in WRF (Cassano et al., 2011). $CO_2$ is set to 290 ppm for a glacial inception scenario (Petit et al., 1999; Vettoretti and Peltier, 2004). The boundary layer is simulated with the Mellor-Yamada-Janic scheme (MYJ, Janjic, 1994). Lateral boundary conditions are prescribed at the edges of the outer domain, yet no nudging to observations is used in the domain interior, as we want to allow the atmosphere to respond to land surface changes. Such spectral nudging is often used in regional models to eliminate artificial circulation patterns that develop in spite of the prescribed boundary conditions (Glisan et al., 2013). Comparing our control run to ECMWF, we find (supplementary Fig. 1) non-negligible deviations in geopotential height. Yet we show below that the anomalous circulation that develops in response to ice growth in our simulations is likely unrelated to these anomalies, as they develop gradually in response to the growing ice height. Note that these physics are the same as those used in our previous paper (Birch et al., 2017), except now we use the Grell convective scheme (Grell and Dévényi, 2002), which has been shown to give good results in present day (Hines et al., 2011) and LGM (Bromwich et al., 2005) Arctic climates.

We use realistic present day land type from the WRF database as initial conditions for the ice extent on Baffin Island. Over these land grid points classified as ice, a snow water equivalent (SWE) of $5000 \, \mathrm{kg \, m^{-2}}$ is prescribed — sufficient for maintaining snow cover for the duration of our runs and allowing us to capture a representative yearly mass balance for the chosen climatology. Greenland is initialized with present day ice cover and elevation based on studies finding little influence of Greenland on the build up of the Laurentide ice sheet (Kubatzki et al., 2006).

We first run a present-day meteorology simulation forced by ECMWF boundary conditions, representative of average and cold atmospheric conditions, also simulated by Birch et al. (2017). With the above WRF configuration, we find a reasonable reproduction of the Arctic atmospheric state when we look at the full fields (Supplementary Figure 1). While there are deviations in geopotential and temperature between WRF and ECMWF, the differences in geopotential have been seen in other Arctic studies (Glisan and Gutowski, 2014). The temperature bias occurs from differences in the snow cover between the models. For instance, ECMWF has been noted to have too large of a snow-covered area on coasts (Drusch et al., 2004), promoting cooling on Baffin Island which is mostly coast at the reanalysis resolution. Also June is the cause of the large temperature bias because ECMWF still has snow on the ground, associated with delayed snow melt (Dutra et al., 2010), while the snow in WRF melts earlier. July and August only differ in temperature by about 1 degree, when the snow cover in ECMWF and WRF are more similar. With the warm temperature bias, WRF will likely not cause an artificial inception, which making it more suitable than models with a cold bias. Furthermore, ECMWF is known to have a cold bias (Bromwich et al., 2002b; Screen and Simmonds, 2011) and a significant low geopotential height in the central Arctic, of the same magnitude as the deviation we see in Supplementary Figure 1. Therefore, with WRF having the opposite biases, the Arctic climate simulated is not abnormal. Our WRF simulation matches observations of precipitation on the coast of Baffin Island, important for the surface mass balance, of about 300 mm per year (Zdanowicz et al., 2012). In any case, the warm bias in our WRF control run does not prevent the circulation from bringing relatively cold and wet conditions to Baffin Island (Bromwich et al., 2002a). Previous works also proceeded to look at climate sensitivity by comparing model experiments in spite of biases in the control run Porter et al. (2012), as we do here, and we therefore do not believe the biases would have a significant impact on the sensitivity results here.

For our main set of glacial inception simulations, the incoming solar radiation is set to values consistent with 115 kya; the process of modifying the insolation in WRF based on the orbital parameters is detailed in the Appendix of Birch et al. (2017). The conditions most conducive to snow accumulation are from October 1985 to October 1986. These cold meteorology boundary conditions caused the present day ice cover to expand when combined with 115 kya insolation, in the study of Birch et al. (2017). Supplementary Fig. 3 shows cumulative mass-balance for present-day average conditions, present-day cold meteorology conditions, and cold meteorology plus 115 kya orbital forcing. The combination of moderately cool temperatures and reduced summer insolation leads to net snow accumulation, hinting that inception might be viable given sufficiently positive regional feedbacks. A longer averaging of the forcing, as obtained from a climate model, would have some advantages. However, given the persistent difficulties in simulating glacial inception using such GCMs, our goal here is to minimize the introduction of biases from global climate simulations. We therefore prefer to choose the necessary cold and wet meteorology from the observed modern climate (reanalysis). We cannot explain how these ideal circulation patterns occur during inception,

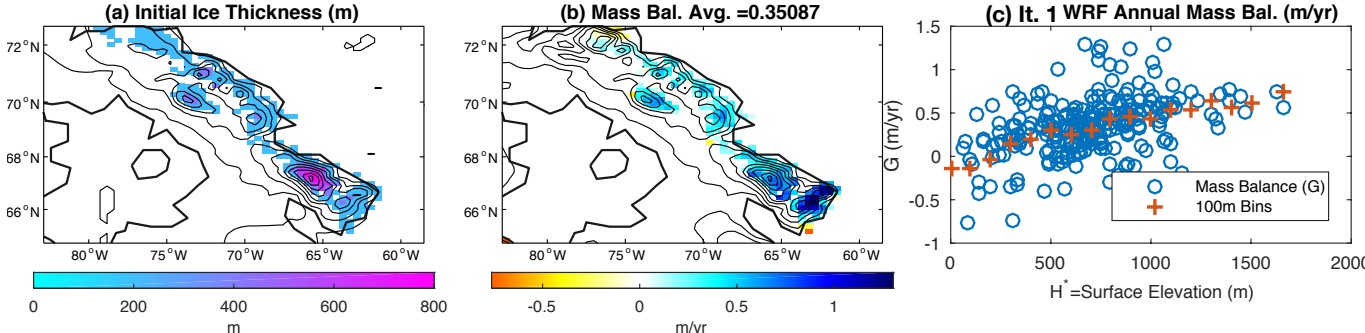

**Figure 1.** Ice thickness and Mass Balance for Ice Sheet Initialization: (a) Ice thickness was adapted from data collected by the NASA Operation IceBridge Mission (Kurtz and Farrell, 2011) with extent set by land use type in WRF. (b) Annual Mass Balance from WRF Iteration 1 Run.(c) Scatter plot of mass balances vs. elevation with a binned regression visualization

but a future study with a GCM might carefully explore this. We do note that forcing with a single-year meteorology may bias the ice growth patterns if that year is not representative of longer-term cold conditions.

## 2.1    Asynchronous Coupling Procedure

We begin our WRF-ice flow coupled model experiment with a WRF simulation that uses meteorology from October 1985
to October 1986 (Iteration 1). WRF is run for 5 years, but we only analyze the last 4 years for an average climatology. Accumulation occurs over areas initialized with ice cover at high elevations, and net mass balance is positive, allowing ice growth (Fig. 1a). The Iteration 1 simulation agrees with our previous 4 km simulation of the Penny Ice Cap (Birch et al., 2017): Milankovitch forcing and cold meteorology cause the positive mass balance needed for the ice cap to expand. This mass balance is then adapted for use in the ice model.

For the first iteration of the ice model, initial ice thickness is adapted from present day ice cover from the NASA Operation IceBridge Mission (Fig. 1a). We use the surface elevation from the 20km resolution Iteration 1 WRF domain. The mass balance needed to force the ice model is calculated from the WRF simulation. Figure 1b depicts the mass balance results from the Iteration 1 WRF simulation with cold meteorology, 115 kya insolation, and present day ice cover. The mass balance over the ice caps is generally positive, except on the edges of the ice cap where there is net ablation. Over the rest of the domain, WRF
cannot calculate a meaningful annual mass balance because all snowfall received melts off during the summer. Thus, to have a mass balance in ice free grid cells, we characterize the mass balance as a function of elevation ($G(H^*)$). This characterization involves sorting the domain by surface elevation and forming bins every 100 m, and allowing the mass balance during the ice model run to be updated at each grid point as the surface elevation and ice thickness change every time step. The average the mass balance in each bin is depicted by the purple crosses in Fig. 1c. Note the plateau in the mass balance at high elevations,
which is expected from the elevation desert effect.

For our asynchronous coupling experiment, the ice model is run for 500 years, as any longer has been shown to inhibit ice growth artificially (Herrington and Poulsen, 2011). We find 500 years sufficient to grow ice while maintaining computationally efficiency. After 500 years, the ice model has calculated a new ice extent and surface elevation (Fig. 2a) with which to force WRF. We import the surface elevation directly into the 20 km resolution WRF domain and use the ice extent to update the

land type, i.e., the expansion of the ice cap. Much like our Iteration 1 simulation, over areas classified as ice, we prescribe a snow depth of 5000 kg m$^{-2}$. The surface elevation and land type of the 100 km resolution parent domain are adjusted through interpolation of the 20km resolution surface elevation and land type. Soil temperatures from the previous WRF run are used to initialize the next WRF iteration. In our simulations, soil temperatures take a year to equilibrate, leading us to agree with de Noblet et al. (1996) that present day soil temperatures are too warm. Again using 115 kya insolation and cold meteorology,

WRF is run for 5 years to characterize the mass balance on this new ice cap. This new run is the start of the next iteration. Thus, each iteration begins with a WRF run and ends after the 500-year ice model integration.

## 3    Results

In the context of glacial inception, expanding ice cover interacts with the atmosphere, causing changes to both the regional climate and atmospheric circulation near Baffin Island. We first examine the progression of the WRF-ice model iterations,

looking at the expansion of ice cover and the resultant changes in mass balance. Then, in Section 3.1, the changes in mass balance are explained by exploring changes in temperature, ablation, precipitation, and cloud cover. In section 3.2, we take a closer look at the regional circulation. Additional sensitivity simulations with the ice sheet model further clarify climate factors that could lead to Baffin Island glaciation in Section 3.5.

The progression of WRF-ice model iterations (5 years of WRF and 500 years of the ice model) is laid out in Figure 2. The

ice thickness calculated by the ice model and then used in WRF is depicted in the left column, and the resulting annual mass balance from WRF in the right column. Also included in the right column is the spatial average of mass balance on the ice cap (defined as all grid cells with ice land cover type). Starting with the top left, we show the ice thickness resulting from running the ice model for 500 years using the Iteration 1 WRF mass balance. This new ice thickness, ice extent, and surface elevation computed by the ice model are used as inputs to the next WRF simulations. Then WRF is run for 5 years with the

mass balance presented in the right column. After the second iteration of WRF, we bin the mass balance by elevation, and force the ice model with the new updated mass balance. A new ice cover is computed (left row, second column) and used as an input in WRF. Subsequent iterations pictured follow the same procedure.

The second iteration has the greatest ice extent, which is due to the large positive Iteration 1 mass balance. In the third iteration, the ice retreats from its margins, though there is still growth in the highest parts of the domain. Additional iterations

show that the ice cap coalsceses and increases in thickness to nearly 2000 m, but expands slowly into new areas. After 5 kyrs, the ice accumulation on Baffin Island would only lower global sea level by 0.6 m. Although the average mass balance does generally increase with time, the rest of our analysis focuses on why the ice does not expand quickly.

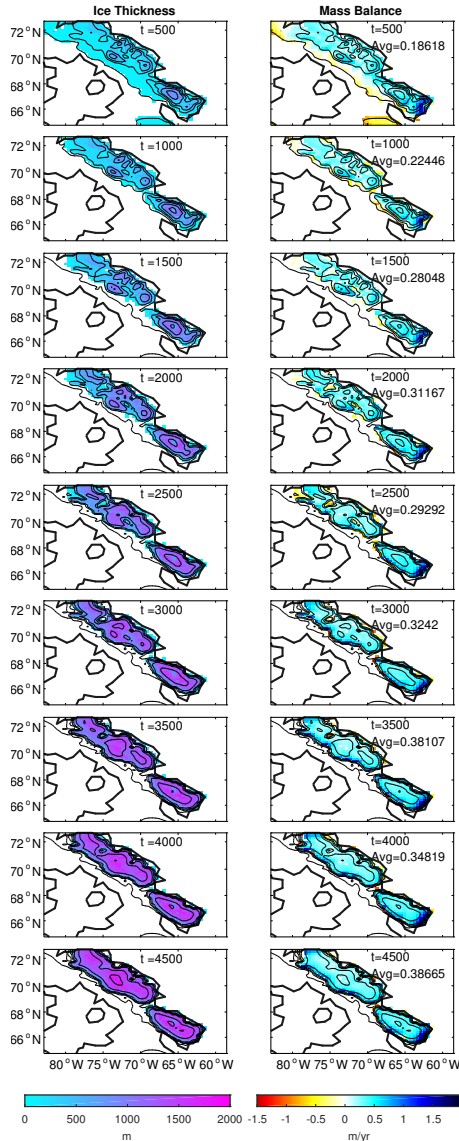

**Figure 2.** Ice growth due to WRF-ice model iterations. Left column is the ice thickness from the ice sheet model. Right column is the mass balance from WRF. The average mass balance over ice-covered areas for each iteration is included in the right column.

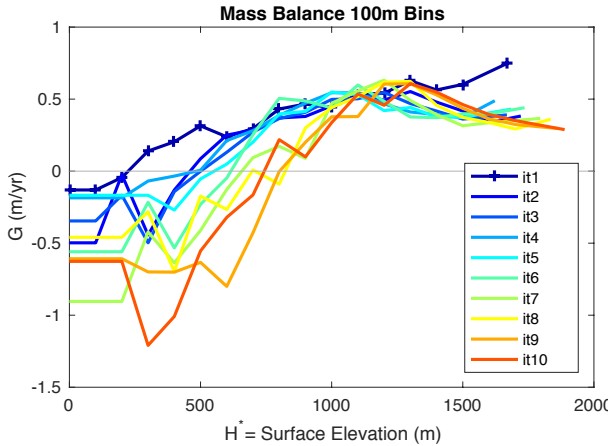

**Figure 3.** Characterizing annual mass balance vs. surface elevation. The domain is sorted into 100 m bins and within each bin the mass balance is averaged. Elevations falling within the range of the bin have the associated mass balance.

The binned mass balance used in the ice model is plotted in Figure 3 with annual mass balance on the y-axis and surface elevation on the x-axis. The Iteration 1 WRF run is the "+" blue line with the highest mass balance, particularly at the lowest and highest elevations. In subsequent WRF runs, the mass balance up to 1000 m becomes more negative at low elevations, which we find surprising. One might expect the equilibrium line to shift downslope as a result of regional cooling from the

initial expansion of the ice cover, leading to a more positive mass balance. However, plotting the binned mass balance for all iterations shows a general trend of increasingly more negative mass balances at low elevations, with the equilibrium line altitude shifting upwards by several hundred meters over the course of ten iterations. As a result, the ice caps coalesce and thicken at high elevations, but stagnate and fail to expand horizontally. The mass balance at highest elevations from all iterations also illustrate desertification, or the mass balance becoming less positive over time as the ice cap increases in height and its top

surface flattens.

Lateral expansion is slow because the lower terrain on Baffin Island is not strongly sloped. Thus, ice must build up to a substantial height before the spread of ice outweighs ablation at the ice edge. For example, with flat basal topography, calculations based on equations 1 and 2 suggest that a 0.5 m/yr of ice thickness increase due to ice flow in a grid cell with no ice requires an adjacent grid cell ice thickness around 500 m. This approximate ice thickness on the margins occurs in our

simulations as depicted in Figure 2.

### 3.1   Mechanisms of Changes in Mass Balance

A closer examination of temperature, melting, and precipitation can help us how the ice cap growth causes the changes to mass balance in the WRF simulations. We begin with the 2 m temperature on Baffin Island during the summer (June, July, August) when melting occurs. The differences in the temperature between the Iteration 1 run and subsequent iterations are

pictured in Figure 4. In the Iteration 1 simulation, we note that most of the ice-covered land remains below freezing during

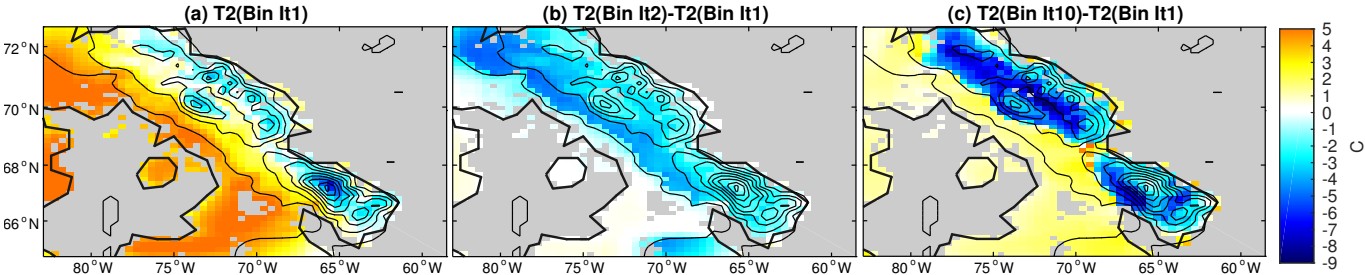

**Figure 4.** The Summer 2m Temperature in select asynchronous coupling iterations: (a) The temperature in Iteration 1 WRF. Differences in temperature between Iteration 1 and (b) second and (c) last iterations.

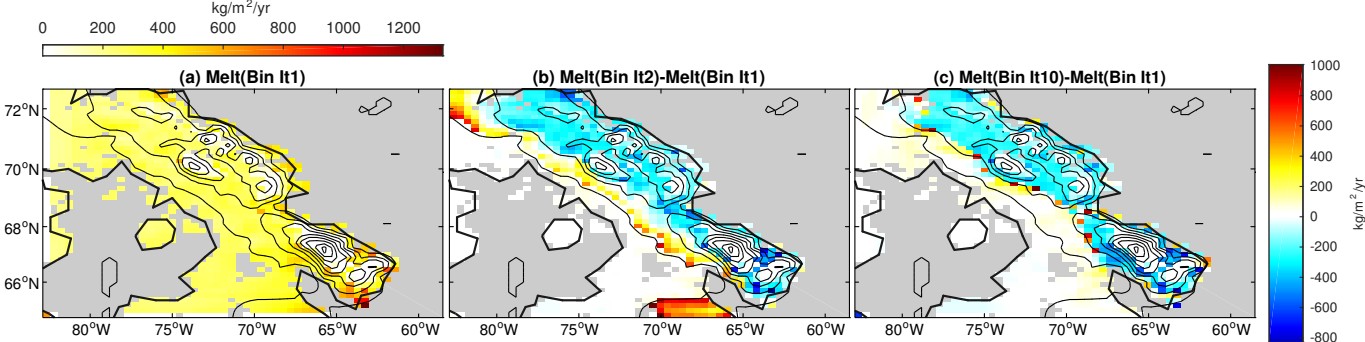

**Figure 5.** Melting in WRF in select iterations: (a) melting in the Iteration 1 WRF run. Differences in melting between Iterations 1 and (b) 2 and (c) last iterations.

the summer (blue). At the edge of the ice cap, temperatures generally remain near zero (white), and the low elevation tundra is above freezing (red). The largest expansion of ice cover occurs in the second iteration as noted above, which dramatically cools much of the region mostly due to the presence of snow and the albedo change. Also note that the upper part of the ice cap also cools, likely due to increasing elevation, since 500 years of ice growth increased the surface elevation by 200 m. In

5  the third iteration, the ice has retreated somewhat, which makes the low elevations warmer than in the second iteration. The integrated mass balance becomes more positive as seen in Figure 2. As iterations of WRF and the ice model continue, we find that the area covered in ice cools, but the low-elevation tundra area surrounding the ice cap warms. The lapse rate effect due to increasing surface elevation progressively cools the ice cap, and below we explore why the low-elevation tundra warms in successive iterations.

10    The mass balance is made up of ablation and precipitation. When precipitation is larger than ablation, the ice cap grows. We are interested in how both metrics change over the iterations of the coupled WRF-ice sheet models. We start by examining the melting over the summer, and how it relates to the temperature discussed above. Figure 5a depicts the yearly melt in Iteration 1 simulation. In general, the high elevation parts of the region experience no melting during the summer, due to their sub-freezing

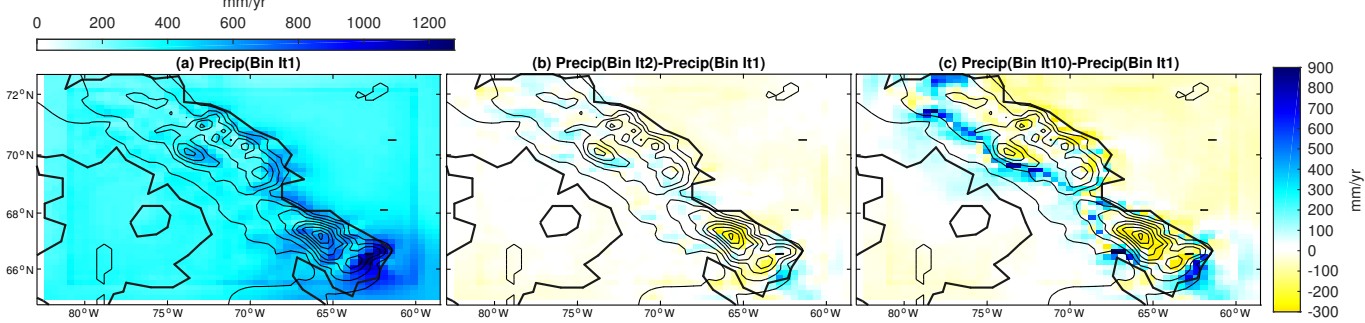

**Figure 6.** Annual precipitation in WRF in select iterations: (a) The precipitation in the Iteration 1 WRF run. Precipitation differences between Iteration 1 and (b) 2 and (c) last.

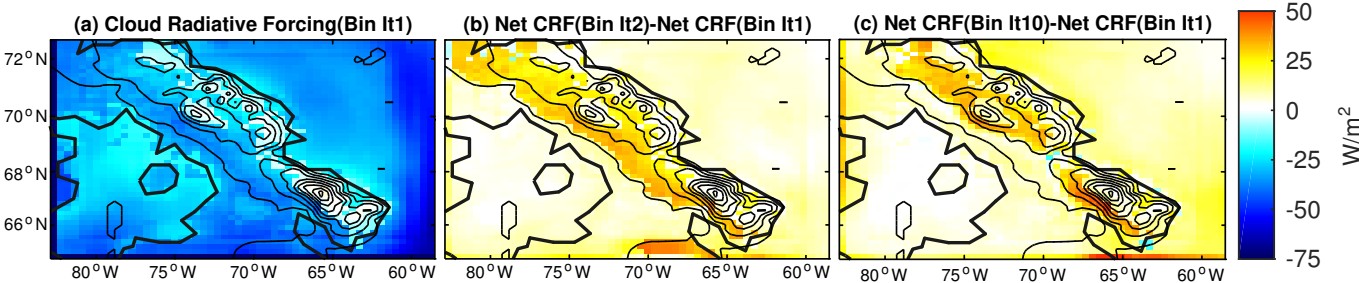

**Figure 7.** The Summer Cloud Radiative Forcing (CRF) in select iterations: (a) The CRF in the Iteration 1 WRF run. Differences in CRF in (b) second and (c) last iterations.

temperatures. As the ice cover expands, the highest values of melting are located near the edges of the ice cap. As expected, melting decreases as the surface elevation increases.

Next, we look at the precipitation component of the mass balance. In this region, precipitation increases with elevation due to the orography (Figure 6). For glacial inception, continued snowfall is critical in addition to low ablation. Figures 6b

5   and c depict the differences in precipitation between the Iteration 1 simulation and ice model iterations. We see an elevation desert effect that reduces precipitation by 200 mm year$^{-1}$, or $\sim 20\%$ of the annual precipitation. The highest elevations receive less precipitation as the iterations go on. However, at the edges of the ice cap, precipitation increases, because the change in elevation and surface slope triggers orographic precipitation there. It is interesting that precipitation increases with ice growth on the Barnes region of the ice cap, which may be a necessary condition for glacial inception to occur. The accumulation

10  increases on the western side of Baffin Island, which is also the windward side and as noted by Roe and Lindzen (2001b), a characteristic of the inception process. This general increase in precipitation near the margins agrees with the maps of the mass balance (Fig. 2) indicating the greatest mass balances do not occur on the peaks of the island, but at about 1200 m elevation.

Finally, we are interested in the role that clouds may play in glacial inception. Jochum et al. (2012) and Birch et al. (2017) found clouds to be negative feedback during glacial inception. When insolation is changed from present day to that appropriate

for the time of glacial inception at 115 kya, Jochum et al. (2012) noted that less low clouds form during glacial inception, while Birch et al. (2017) found that cloud properties change such that relative to present day, less shortwave is blocked from the surface. Our objective here is to examine this feedback during the integration of our asynchronously coupled model. In Figure 7, the net CRF in the Iteration 1 simulation is negative during the summer because the clouds block shortwave radiation from reaching the surface. However, as the ice cover expands, the clouds block less shortwave radiation, illustrated by the positive net shortwave CRF difference between later iterations and the Iteration 1 run: the Iteration 1 simulation net CRF is more negative in all parts of the domain. The albedo of increasing snow cover accounts for less negative CRF over newly glaciated areas, but the change to the shortwave CRF in the rest of the domain is due to changes in cloud cover and properties.

## 3.2 Interaction of Baffin Island Inception and Atmospheric Circulation

We find the significant warming over Baffin Island to be surprising, as one might expect the expanded ice cover over the peaks of Baffin Island to lead to cooling (which would amplify the positive mass balance and lead to a larger ice sheet). We now analyze the warming by looking at the regional geopotential height field, temperature, and winds in the outer domain. First, the geopotential height and winds at 500 mbar for the Iteration 1 simulation during the summer are depicted in Fig. 8. The winds over Baffin Island generally comes from the northwest with a low pressure trough covering the whole island. A weak cyclonic flow also exists over the island and Baffin Bay. Next we examine the subsequent iterations and their anomalies in the winds and geopotential height at 500 mbar in Figs. 8b-c. The pressure increases south of Baffin Island while it decreases over Baffin Bay to the north, and an anomalous anticyclone forms over the Hudson Bay region. It seems that the larger ice caps on Baffin Island alter the circulation, and we will examine this hypothesis below.

The 2m temperature response in the outer domain (Figure 9) is consistent with the changes in geopotential height. The Iteration 1 simulation is depicted in the upper left with subsequent panels showing differences in the following iterations. On the ice caps of Baffin Island, the temperature generally grows colder because the elevation has increased, and newly glaciated areas cool due to both the presence of ice and the elevation increase. As iterations progress, the non-glaciated points of the island warm, which is also seen in the inner domain. Additionally, warming occurs outside of Baffin Island in the Hudson Bay region. These areas of warming, coinciding with geopotential height anomalies, appear to be caused by the circulation response, with warm southerly advection by the anomalous anticyclone.

This warming appears to be a negative feedback on ice sheet growth, and it explains the mass balances becoming more negative at low elevations on Baffin Island through subsequent iterations (Fig. 3). With a region so close to freezing during the summer even a degree warming can prevent perennial snow from occurring during the critical inception period. As a caveat, we note that with the elevation in the coarser outer domain being generally lower than the 20km resolution domain, the temperatures on Baffin Island are much warmer than is likely realistic. For instance we do not see any summer temperatures significantly below freezing as found in the inner domain with more realistic topography.

It is known that regional models can develop artificial anomalous circulations (Glisan et al., 2013) and it is important to verify that the negative feedback identified here is not strongly affected by such artifacts. Fig. 10 shows two scatter plots of the temperature anomaly and the geopotential anomaly as function of ice height for iterations 2-10. The anomalous circulation

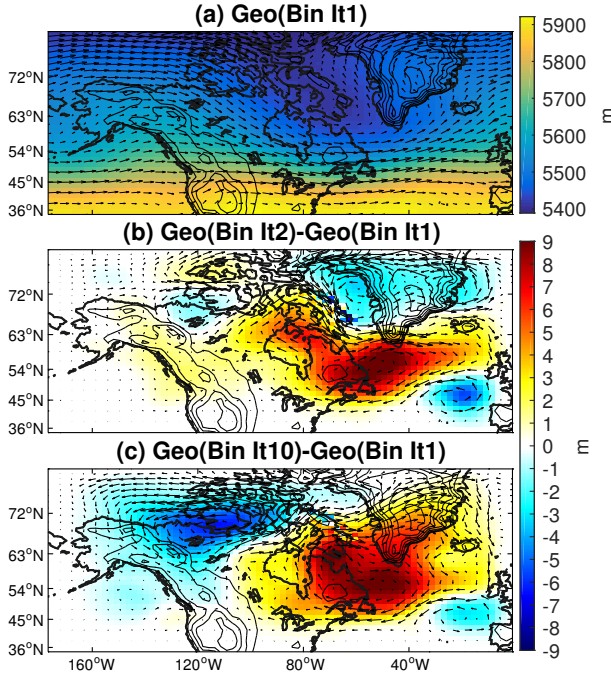

**Figure 8.** Changes in 500-mbar geopotential height in the Parent Domain in select iterations with wind vectors. Wind speeds are 6 m s$^{-1}$ over Baffin Island. (a) The 500-mbar geopotential height for the Iteration 1 simulation with average largescale flow during the summer (b)-(c) 500-mbar geopotential hegith differences during the summer between Iteration 1 and iterations. Wind anomalies of 1 m s$^{-1}$ are overlaid.

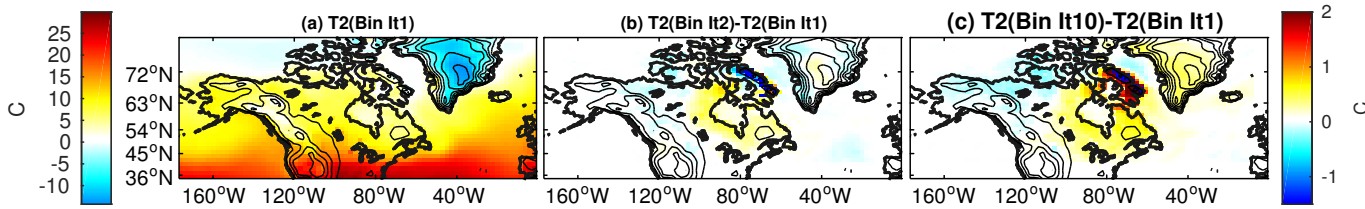

**Figure 9.** Summer 2m Temperature in the Parent Domain in select iterations: (a) The 2m Temperature for the control simulation (b)-(c) 2m Temperature differences during the summer between control and iterations.

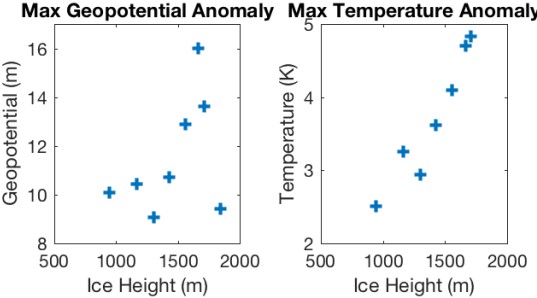

**Figure 10.** Scatter plots of summer (JJA) maximum (a) temperature anomaly and (b) geopotential height anomaly over Baffin Island vs. ice elevation in each iteration.

and heating develops progressively with the ice growth, suggesting that the anomalous circulation is driven by the changing ice topography and is likely not an artifact. We further investigate the effect of topography in the following section. The geopotential anomaly increases with ice height, with the exception of the last iteration when the topography undergoes a new large area of ice appears. Given that the temperature anomaly is still strong in this last iteration, the feedback is still valid
through this last iteration. The response we find is consistent with previous studies who noted that the size of the mountain is critical to the orographic response (Chen and Lin, 2005; Cook and Held, 1988) and slight changes to topography can cause significant changes (Roe and Lindzen, 2001a). We speculate that the physical mechanism underlying the topography-induced anticyclone might be related to mixing of shallow-water potential vorticity (PV) by eddies near a localized high point in a weak background flow. Lateral eddy diffusion near a high point would stir high-PV air (due to lower thicknesses) away from the high
terrain. The anti-cyclone is created by the influx of lower-PV air from the surrounding environment. The anticyclone would theoretically increase in strength as the surface height increased, as seen in Fig. 10. This assumes eddy mixing to be unchanged, which may explain the weakened anomaly in Iteration 10. The anomaly seen here differs from the more commonly considered atmospheric situation where advection of PV by the mean flow dominates horizontal mixing. A more thorough consideration of this simplified problem, including eddy mixing, mean flow, and planetary vorticity gradients, is left as a subject for future
work.

### 3.3 The of Impact Baffin Island Ice Elevation on Regional Circulation

Next, we diagnose the roles of ice and topography on circulation changes, by setting up sensitivity simulations based on the second iteration, which has changes in ice extent and surface elevation (Fig.2). For our "No H" scenario, we keep the surface elevation the same as the Iteration 1 simulation, but use the same ice extent as in the second iteration. For an "H Only" scenario,
we remove all ice, changing the land type classification to tundra. We then set the surface elevation to the ice height in the second iteration. These two additional simulations were run for 5 years with cold meteorology and 115 kya insolation and the analysis focuses on the last 4 years.

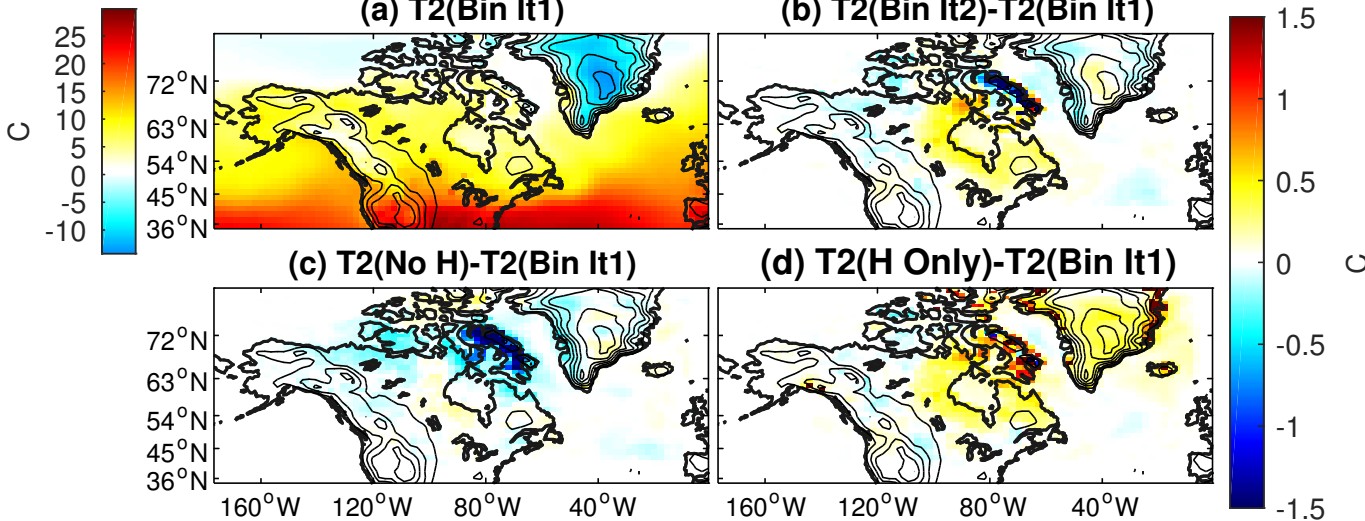

**Figure 11.** Changes in 2m Summer Temperature in the Parent Domain for topography and ice sensitivity simulations. (a) The 2m Temperature for the Iteration 1 simulation with averaged during the summer (JJA) (b) differences during the summer between Iterations 1 and 2. (c) differences between the Iteration 1 and No H. (d) differences between Iteration 1 and H Only.

In Figure 11, we examine the temperature of the outer domain for 4 experiments. Again, the Iteration 1 simulation summer temperature and difference from the second iteration (having both ice and elevation changes) are shown, but we also include the "No H" and "H Only", in c and d respectively. With "No H", a significant regional cooling occurs, encompassing both Baffin Island and the Hudson Bay region, which is the pattern we had anticipated to cause rapid ice growth. However, with 5 "H Only", warming on both Baffin Island and around Hudson Bay occurs. The geopotential height anomalies presented in supplementary Figure 2 agree with the temperature patterns seen here. Thus, the cause of the negative feedback due to large scale anticyclonic circulation and warming in our simulations is the change in topography. An interesting aspect of these last two simulations ("No H" and "H Only") is that if we sum the difference between their response and the Iteration 1 simulation, the result is very close to the difference between iteration two and the Iteration 1 run, as seen in Fig. 12. This indicates that the 10 responses to the ice topography and ice land type (albedo and other surface properties) are linear.

### 3.4 Glacial Inception with Changing Orbital Parameters

The WRF-ice model iterations performed previously used the minimum insolation from 115kya, which is useful for diagnosing the relationship between the ice cover and atmosphere. However, that is obviously not realistic. Integrated insolation increased after its minimum at 115 kya, impacting temperature and glacial mass balance. We use the procedure detailed in the Appendix 15 of Birch et al. (2017) to set the insolation in WRF for 115-111 kya. After each WRF run, the ice model is integrated for 500 years using the same mass balance procedure as detailed above. This set of simulations explores the mass balance that results from a combination of ice flow and insolation changes.

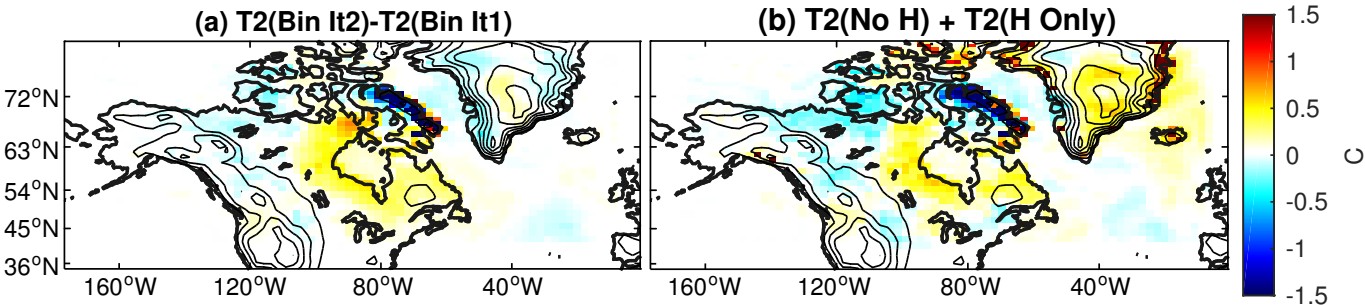

**Figure 12.** (a) Differences during the summer 2m temperature between the first and second iterations. (b) Sum of temperature differences from "No H" and "H Only" scenarios, implying a linear response due to changes in ice cover and surface elevation

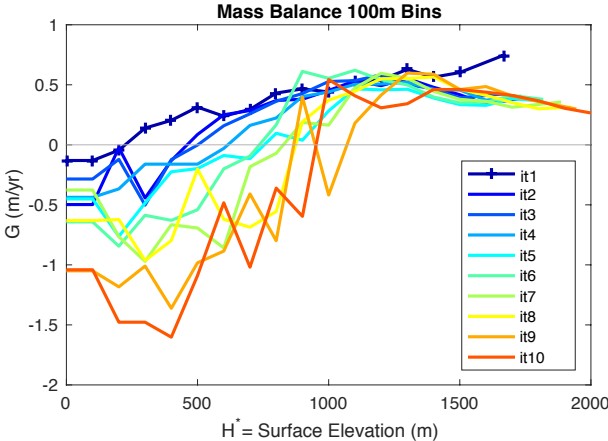

**Figure 13.** The Mass Balance Binned by Elevation used in simulations with realistic insolation from 115 kya to 111 kya, where the first and second iterations use 115 kya insolation, third and fourth use 114 kya insolation, and so forth to iteration 10 with 111 kya.

Figure 13 presents the mass balances calculated from WRF and used in each iteration of the ice model. Note the Iteration 1 simulation and 2nd iteration plotted here are the same as presented in Figure 3 because the change in insolation occurs at 114 kya, which is the third iteration in Fig. 13. The elevation desert effect is again observed and the mass balances become more negative at low elevations, hindering lateral growth. Thus, these iterations progress in a similar manner as the case with constant
5   insolation in that the lateral growth is slow though the height of the ice sheets does grow, but the high elevation plateaus can be 200 m lower than the previous set of simulations with constant 115 kya. In this set of simulations with realistic insolation the mass balances at low elevations are more negative, explaining the slower growth (Fig. 13).

During the summer on Baffin Island, there is again cooling from the presence of ice, but the unglaciated parts of Baffin Island warm considerably. As the summer insolation increases, the summer temperature of all of North America, not just the
10   Hudson Bay region, increases as well. When we look at the precipitation in the domain, we observe a similar pattern to the previous set of simulations. Accumulation increases on the edges of the ice cap due to orographic precipitation. Then at the

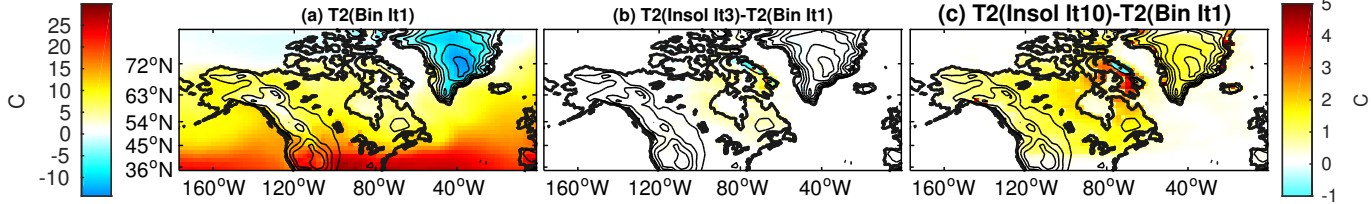

**Figure 14.** The surface temperature in the Iteration 1 run, and differences for successive iterations.

top of the ice cap, the elevation desert effect kicks in. The ice cap continually receives less precipitation than in the Iteration 1 simulation. The increases in insolation causes continental-scale warming, and when combined with topography changes, the warming on Baffin Island can be over 5 K (Figure 14). Thus, the realistic insolation changes cause further warming during inception, which makes the occurrence of inception even more difficult to explain and simulate.

## 3.5 Sensitivity Analysis of the Ice Flow Model

The combination of WRF and the ice flow model does not indicate ice sheets would grow to cover all of Baffin Island, which would likely be necessary to explain the sea-level drop found in the paleorecord. The change in ice surface elevation actually causes a negative feedback on inception by affecting the large scale circulation: temperatures around the ice increase inhibiting ice expansion. This result suggests a closer examination of the ice model and its sensitivities. In particular, we vary the creep parameter $A$ to see if rapid ice expansion is possible with different constant choices. We also explore how a successful inception - matching paleorecord estimates - could occur from our simulated mass balance function $G(H^*)$ and limiting the calving loss of ice in Hudson Bay. The ice model in these sensitivity experiments is now run for 10kyrs, unlike our previous asynchronous simulations that ran for 5 kyrs total.

First, we identify if we have simulated a mass balance sufficient for a successful inception. From Figure 3, we note that the mass balance in the Iteration 1 simulation is the most positive at high elevations and least negative at low elevations. All subsequent WRF runs simulate a negative mass balance at low surface elevations. Using the Iteration 1 mass balance, the ice model is integrated over 10 kyrs. Of course, this situation is not realistic as our set of simulations show changes in mass balance due to circulation changes as the ice sheet grows. We prescribe the mass balance only on Baffin Island, which was the location of our inner domain, and this causes Baffin Island to be mostly covered by ice, corresponding to 1.75 m sea level drop over 10 kyrs. Ice model simulations forced with the mass balances from iterations 2-10 never reach the western coast of Baffin Island in 10kyrs.

As noted above, we choose the creep parameter $(A(T) = A(-10°C))$ based on Cuffey and Paterson (2010). We now test the sensitivity of the ice flow when $A(T)$ is evaluated at warmer and colder temperatures, or A(0°C) and A(-15°C) respectively. For these simulations, the mass balance used is from the Iteration 1 WRF simulations, and when run for 500 years, the ice model changes are minor. We confirm with WRF simulations that these slight changes in ice elevation from A( 0°C) and A( -15°) do not significantly affect the mass balance in the second iteration; the ice still retreats. Thus, we want to examine how

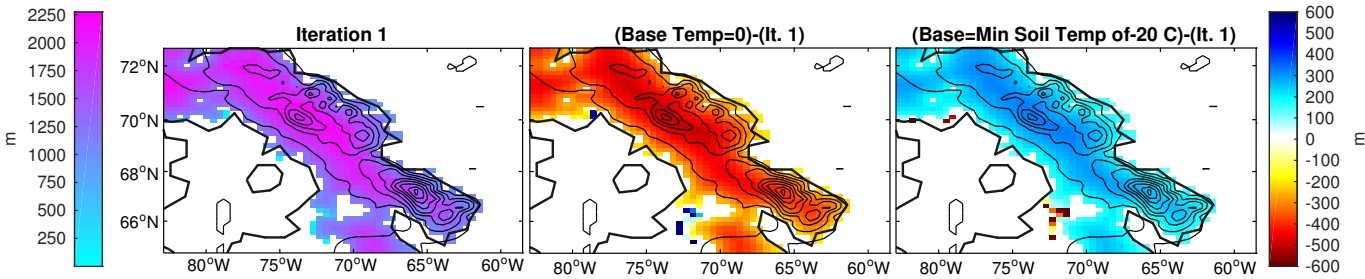

**Figure 15.** Sensitivity of the Ice Model to choice of constant creep parameter $A(T)$, which is related to the temperature profile within the ice as detailed by Cuffey and Paterson (2010). (a) $A$ evaluated at -10°C or A( -10°)C. (b) A( 0°C). (c) A( -15°)C

the creep parameter affects ice flow on longer timescales, which is why we integrate the ice model in time for 10 kyrs. We expect that a warmer temperature and thus larger $A$ would allow the ice to spread more easily, leading to larger ice cover.Figure 15 shows that changing the creep parameter can increase or decrease the thickness of a Baffin island ice cap by about 20%, but does not alter the ice extent much after 10 kyr. On average, $A(-10°C)$ causes the elevation to be 200 m higher than $A(0°C)$.

More calving occurs with $A(0°C)$ ice reaches the coast sooner. We also examine the impact of temperatures at about $-15°C$ (Fig. 15c), which Marshall et al. (2002) determined to be the basal temperature of the initial ice sheet growth. As expected, the ice flows more slowly and accumulates at high elevations. These sensitivities to the creep parameter, however, are small compared to the sensitivity of ice extent and thickness to the mass balance G(H*) as inferred from successive iterations of WRF.

We also question what would happen if we apply the Iteration 1 mass balance to a larger area, outside of Baffin Island and our chosen domain. Based on the temperature and precipitation patterns of the outer domain, the other side of Hudson Bay could have a similar mass balance, especially if the domain were resolved at a higher resolution. We do not include the Torngat Mountains on mainland Canada just to the south of Baffin Island, as this area appears to receive less precipitation and experience higher temperatures. This agrees with observations on the lack of ice today, indicating the possibility of later

glacierization (Andrews and Miller, 1972). The ice accumulation with the positive mass balance from our Iteration 1 WRF simulation on mainland Canada causes an equivalent sea level drop of 2.8 m in 10 kyrs.

    The previous simulations assume immediate calving over any ocean points, which makes sense for the kilometer deep Baffin Bay. The topography on the eastern side of Baffin Island is steep, possibly preventing ice shelves from forming. However, Hudson Bay is only about 200 m deep. We turn off immediate calving for this shallow water body, and instead prescribe a

negative mass balance. We extend the regression $G(H)$ linearly to the bathymetry of Hudson Bay. This allows for ice to flow slowly into a larger area surrounding Baffin Island, creating a fairly large ice sheet and corresponding to a sea level drop of 6.6 m. This last scenario gets close to more recent constraints of sea level drop from modeling simulations at around 100 kya of 10m (Creveling et al., 2017). This growth of the ice sheet corresponds a successful inception, though still on the slower side of observational constraints, suggesting that the negative feedback due to the response of the large scale circulation to expanding

ice cover is what prevents our asynchronously coupled model from achieving a successful inception.

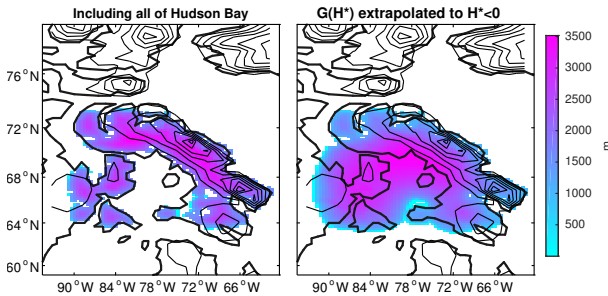

**Figure 16.** The result of running the ice model for 10kyrs with the maximum mass balance, found in the first WRF iteration. (a) Allowing a positive mass balance on mainland Canada, surrounding all of Hudson Bay. (b) G(H*) extrapolated to H*<0 in Hudson Bay, which allows for ice flow instead of immediate calving of ice.

## 4 Discussion

In this paper, we have attempted to simulate the last glacial inception beginning at 115kya with a regional high resolution configuration of WRF, coupled asynchronously with a simple ice flow model. A net positive mass balance occurs in the 1st WRF iteration simulation driven by 115 kya insolation, cold present day (1986) boundary conditions, and present day ice cover. If the mass balance from this 1st iteration was maintained, an ice sheet could cover all of Baffin Island after 10 kyr. However, updating the ice cover and surface elevation in WRF creates a negative feedback from changes in the large scale circulation: summer temperatures over Baffin Island increase due to a regional anticyclonic circulation. This causes the mass balance over Baffin Island to become more negative at low elevations, stagnating further ice expansion in the ice flow model. This is reminiscent to the findings of Herrington and Poulsen (2011) and Gregory et al. (2012) who noted the continued growth of the ice sheet is slowed by changes in large scale circulation that arise from atmosphere-topography interactions. While the studies of Herrington and Poulsen (2011) and Gregory et al. (2012) found the negative feedback slowed ice growth mostly when the ice sheet was fairly developed, of a continental-scale, the negative feedback analyzed here occurs upon inception, as soon as the ice caps over the Baffin Island begins to grow. Previous studies (Herrington and Poulsen, 2011; Gregory et al., 2012) could not resolve this feedback with their coarse resolution GCMs, while our coupled ice sheet - regional atmospheric model enables us to examine it.

The flow over Baffin Island driven by the 1986 boundary conditions is consistently from the northwest during the summer, with weak speeds of about 6 m s$^{-1}$ at 500 mbar. The increase in the ice surface elevation after the 1st iterations of the coupled model causes an anticyclonic response of about 1 m s$^{-1}$ to develop to the south of Baffin Island while a cyclone develops in the north, diverting the winds and bringing in warmer air from the south. This leads to net ablation and arrests the glaciation process. This circulation response is likely dependent on the initial flow (direction and speed) and topographic height (Chen and Lin, 2005; Cook and Held, 1988). Thus, a different meteorology choice might lead to sustained accumulation. Bromwich et al. (2002a) identified exceptionally cold conditions on Baffin Island and their varying large scale circulation patterns. Generally, the cold summer circulation flows from the northwest or north-north-west. We examine the role of meteorology directly using

boundary conditions that differ from those used in our cold meteorology simulations. For this purpose, WRF is driven using 1979 boundary conditions, which is an average year in regards to temperature and precipitation, and the flow over Baffin Island generally comes from the west. WRF simulates a negative mass balance for this year, and the ice sheet retreats to an elevation above 1400 m when the ice model is forced with this mass balance for 500 years. In contrast, the ice cap from our average 1979 meteorology at a 4 km resolution was in approximate equilibrium, leaning towards slight expansion (Birch et al., 2017). Net ablation occurs in the 20 km resolution because of the lower topography. It is possible that during inception that the circulation pattern was different and allowed for both the initial ice expansion and continued cold conditions. WRF did not simulate such a pattern in this study, nor did the few "cold" meteorological conditions in the ERA-Interim reanalysis.

We find significant warming on Baffin Island and the surrounding Hudson Bay region in response to the initial ice growth due to anticyclonic flow. The formation of anticyclones over expanding ice sheets has been noted to allow cold northerly winds to flow over the ice sheet, enhancing inception (Roe and Lindzen, 2001b; Liakka et al., 2012). In particular, small ice sheets cause a wave response that enhances accumulation, while stationary waves induce less cooling as ice sheets increase in size (Liakka and Nilsson, 2010). In contrast, the anticyclone in our simulations forms south of the ice growing on Baffin Island, bringing significant warming even over these relatively small mountain glaciers. Similar to our study, Herrington and Poulsen (2011) found the Hudson Bay warmed in their coupled atmosphere-ice simulations due to anticyclonic flow. Gregory et al. (2012) also found warming on the margins from anticylones and reduced cloudiness. In both of these studies, though, the warming only slowed growth; the regional cooling due to the growing ice sheet kept the ice advancing. We find the heat fluxes into the region by the developing anticyclone circulation to completely overpower any regional cooling from the snow albedo feedback, stagnating the ice sheet growth. We do observe the development of katabatic winds in response to ice sheet growth in our model, possibly contributing to the drying out of the region and decreasing cloud cover, but these winds do not cool the area outside of the glaciers sufficiently to allow for inception.

We do not find a positive feedback that would lead to a more positive mass balance due to the initial ice growth. However, some of our assumptions and modeling choices could be preventing such a positive feedback. For instance, our WRF simulations prescribe the SSTs and sea ice to those of 1986, but during glacial inception the SST could decrease and sea ice extent grow larger (Meissner et al., 2003). Allowing the SST to cool has been shown to induce cooling over land, even if the atmosphere is relatively warm (Yoshimori et al., 2002), while Kageyama and Valdes (2000) noted that storm tracks change with colder SSTs in a manner conducive for inception. In contrast, Stokes (1955) and Gildor and Tziperman (2000, 2001) argued that the oceans change slower than the atmosphere, so the oceans may remain warmer and ice free at the beginning of inception. Goñi et al. (2005) argued that SSTs remained warm until 105 kya, when substantial ice sheets had formed. Cortijo et al. (1994) noted that the northern Atlantic remained close to modern day temperatures through 115 kya until glacial stage 5d around 110 kya. Vimeux et al. (1999) argued that observations support a warm ocean, similar to an interglacial, during inception, though their focus is on the souther hemisphere. Some past GCM studies yielding some degree of glacial inception have also found Atlantic Ocean temperatures to remain similar to present day (Jochum et al., 2012; Otieno et al., 2012).

Oceans may also affect moisture availability, not just heat fluxes. The amount of sea ice can impact accumulation over the ice sheet (Yin and Battisti, 2001; Kubatzki et al., 2006) due to decreasing atmospheric moisture and evaporation from ocean,

according to the temperature precipitation feedback (Källén et al., 1979; Ghil, 1994; Gildor and Tziperman, 2000, 2001; Li et al., 2005). Some studies argued that the cooling during inception did not cause sea ice to extend outside of Baffin Bay until 80kya (Löfverström et al., 2014), meaning the Atlantic remained ice free, which was conducive to high precipitation. Both Khodri et al. (2001) and Wang and Mysak (2002) found that including ocean feedbacks caused increased moisture fluxes

conducive to inception, while Jackson and Broccoli (2003) and Kaspar et al. (2007) noted that storm frequency increased. Including ocean feedbacks could lead to increased moisture fluxes and thus precipitation, or it could lead to cooler temperatures decreasing melting, which could be more important for inception. With present day fluxes prescribed at the boundaries, and the focus of this study being regional feedbacks, we keep SSTs at modern values for consistency, and effectively investigate the "warm" interglacial temperature ocean scenario, which does have support in observations (Ruddiman et al., 1980).

We fix the $CO_2$ concentration to 290 ppm, although $CO_2$ decreased during inception. This decrease has been shown to influence the accumulation of northern hemisphere ice sheets (Ganopolski and Calov, 2011; Vettoretti and Peltier, 2004), accounting for up to 50% more ice (Bonelli et al., 2009). We also do not allow for dynamic vegetation, but on Baffin Island the tree canopy does not exist. Most of these studies dealing with dynamic vegetation refer to the treeline moving southward during glacial inception (Harvey, 1989; Goñi et al., 2005), enhancing ice growth, but on Baffin Island the vegetation is mostly tundra

and small shrubs, which are not an impediment to snow accumulation as trees would be (Otterman et al., 1984; Fréchette et al., 2006). However, the absence of dynamic vegetation means we cannot explore the cooling effects of albedo changes outside of Baffin Island due to vegetation changes. The southward treeline trend and the subsequent albedo impact, though, has been shown to occur later in the inception process (Goñi et al., 2005), justifying our neglect of this feedback.

We choose to drive our model with meteorology conducive to a glaciation inception, and force the model with anomalously

high precipitation and cool temperatures over Baffin Island in 1986. We are assuming that the planetary waves response to the growing Laurentide ice sheet (Wells, 1983) was not significant until the size of the Laurentide was substantial, as has been found in other modeling studies of glacial inception (Otieno et al., 2012). The regional flow induced by these boundary conditions in conjunction with ice growth causes the negative feedback discussed above. The strength of this negative regional feedback that prevents inception from proceeding, suggests that a reorganization of the planetary-scale Northern Hemisphere circulation

could occur as part of the inception process and counter the regional negative feedback. We cannot capture such a change in the general circulation with our regional model and prescribed boundary conditions, indicating a study of non-local processes could complement this study of local Baffin Island feedbacks. For instance, the Laurentide ice sheet and Fennoscandian ice sheet in Europe are thought to have begun growing at the same time, though the Laurentide increased in volume more rapidly (Kleman et al., 2013; Ganopolski et al., 2010). The European ice sheet may have influenced the formation of the Laurentide

(Beghin et al., 2014), but such a feedback is missing in our regional model. Roe and Lindzen (2001b) noted that changes in the stationary wave pattern would affect the downstream climate.

We limit our study to Baffin Island based on geological records of it being the nucleation of the Laurentide, but regional modeling of other Canadian mountain ranges could help identify how easily other areas glaciated, contributing to sea level drop. Growth of ice at high elevations simultaneously throughout Eastern Canada is not impossible, which could all coalesce

into the Laurentide. For instance, various ice sheet models have shown ice beginning to accumulate all over Canada (Andrews

and Mahaffy, 1976; Marshall and Clarke, 1999). While many of these ice modes do not match geological records because they simulate the growth of ice in the West, ice growth on many of the more eastern mountain ranges could perhaps impact the large-scale circulation in a manner conducive to inception.

Though we do find net accumulation in our Iteration 1 simulation using a 20 km resolution, there are inherent issues that come from this resolution. The topography of Baffin Island is very rough, but is greatly smoothed by the 20 km resolution. More importantly, the highest elevation peaks with 20 km resolution is $\sim$ 400 m below the peaks found in realistic Baffin Island topography. Our simulation with average meteorology demonstrates that resolution increases can easily switch the mass balance from net accumulation to net ablation. For cold meteorology, the resulting mass balance is similar to what we found with a 4 km resolution (Birch et al., 2017).

To get an average mass balance to drive the ice sheet model, WRF is run for 5 years; we use the average mass balance from the last 4 years. We find that the land model required a 1 year spin up time, although often a spin up time of more than a year is recommended (Yang et al., 1995; Cosgrove et al., 2003). We tested running the model for longer than 10 years, but the simulations did not deviate much from years 2-5 of our simulations. Deser et al. (2014) showed that multiple ensemble members of an atmospheric model simulation can give different results on a time scale up to 50 years, but this may not be an issue here because we are driving our regional model with ideal inception boundary conditions that constrain the synoptic and planetary-scale flows (Denis et al., 2002).

Using the shallow ice approximation neglects important ice dynamics, in particular it neglects lateral stresses and does not allow for the development of ice streams, which cause rapid ice flow. It has also been suggested that a finer grid size in the ice model could make for more accurate flow, as coarser resolutions have been shown to halt ice flow (Van den Berg et al., 2006). As noted above, lateral expansion only occurs when substantial ice thickness exists in an adjacent grid cell. However, we find that simulations with a 4 km resolution (Supplementary Fig. 6) yield similar ice flow patterns to the 20 km results we present here. We are neglecting basal sliding that could as the ice base melts due to geothermal heat flux (Marshall and Clark, 2002), but the initiating ice sheet was likely frozen at the bed (Marshall et al., 2000), which makes our choice appropriate. Finally, we are not accounting for glacial isostatic adjustment. Herrington and Poulsen (2011) found that the isostatic adjustment was not critical for glacial inception, though including the adjustment revealed a small positive feedback. Therefore, contrary to conventional wisdom, these studies suggest that isostatic adjustment might actually favor glacial inception over Baffin Island, and may even be critical to the process by reducing the strength of negative elevation-circulation feedbacks on ice growth (as in the "No H" sensitivity test). Our binning procedure and the resulting SMB forcing recipe, while simple, are consistent with the way many simple ice sheet models are forced. More importantly, the results of this procedure, showing that ice elevation causes a negative warming feedback, should be robust regardless of how the ice elevation was calculated, as it is a result of the atmospheric model itself. We found that finer bin sizes did not significantly affect the expansion of ice. However, it would be good to test alternative SMB recipes, such as using mean monthly air temperatures and precipitation from the previous WRF integrations, and downscale them to the surface elevation of ice grid points.

Though we find negative feedbacks on the growth of ice sheets due to clouds and circulation preventing inception in our model, high resolution regional modeling still has potential to make progress on the inception problem. We have captured the

mass balance on critical mountainous regions in our Iteration 1 run, which GCMs cannot simulate accurately. When combined with ice flow, the Iteration 1 mass balance causes a significant portion of Baffin Island and the surrounding area to become ice covered. Though the equivalent sea level drop is only 6 m in 10 kyrs, the geological constraints on sea level drop from the Laurentide are not tight, and the calculated sea level drop may be consistent with more recent sea level estimates (Creveling

5 et al., 2017). However, this scenario neglects the ice-elevation feedback on the atmosphere that we found significant. Given the geological importance of Baffin Island on glacial inception, additional work is needed to understand how Baffin Island cooled against two warming influences, the regional anti-cyclone circulation and increasing insolation from 115 kya onward. The warming feedback due to local circulation changes that we find could be mitigated by changes in the pattern of large-scale atmospheric circulation, including stationary and planetary waves. Using a circumpolar regional model could potentially

10 capture the mass balance of the nucleation points of both the Laurentide and Fennoscandian ice sheets at a sufficiently high resolution, allowing an investigation in the downstream interplay of these two ice sheets. Another interesting study could involve exploring the possible impact of a smaller Greenland ice sheet upon exiting the last interglacial. Going a step further to more non-local feedbacks (such as SST changes) influenced by changing orbital parameters, a next step would be simulating the last glacial inception with a GCM and using the results as boundary conditions in a regional model.

## Appendix A: Shallow Ice Model Set-Up

The simple ice sheet model is based off of Oerlemans (1981), where ice flow is represented by a non-linear diffusion equation and the diffusivity contains a constant term called the creep factor. This parameter determines the ice viscosity and depends on the temperature of the ice (Cuffey and Paterson, 2010), meaning it can be denoted $A(T)$ as illustrated in Equation A1:

5 $$A(T) = A_0 e^{-Q/RT} \tag{A1}$$

If $T < -10$C, then $A_0 = 3.61 \times 10^{-13}$ Pa$^{-3}$ s$^{-1}$ and $Q = 60 \times 10^3$ J mol$^{-1}$, but if $T \geq -10$C, then $A_0 = 1.734 \times 10^3$ Pa$^{-3}$ s$^{-1}$ and $Q = 139 \times 10^3$ J mol$^{-1}$.

The surface of the ice is colder than the base, meaning that throughout a real ice sheet temperature, and thus $A(T)$, vary. In 10 fact, the basal temperature often reaches $0°$C for thick enough ice sheets. For an ice model using the shallow ice approximation the standard procedure is to take the vertical average of $A(T)$ (Weertman, 1957; Goodman and Pierrehumbert, 2003) in Equation A2:

$$A(T) = \frac{1}{h} A(T(z)) dz \tag{A2}$$

15 Since $A(T(z))$ varies by height, we need a vertical temperature profile within the ice sheet. We calculate it using the average temperature of the ice surface from WRF and a geothermal heat flux of 55 W m$^{-1}$, assuming a linear vertical temperature profile. We then evaluate $A(T)$ for this temperature profile as found in Equation A and take the vertical average. Typically, this vertical average is influenced by $A(T)$ at the warmer temperatures at the base of the ice sheet, since $A(T)$ varies exponentially with temperature. We calculate $A(T) = 3.5 \times 10^{-25}$ Pa$^{-3}$s$^{-1}$, which is the same as A($-10°$C) from Cuffey and Paterson 20 (2010). We find that using Robin's solution (Cuffey and Paterson, 2010) instead of a linear profile confirms that the basal temperature is below freezing, consistent with our shallow ice model formulation. The basal temperature we use is not far from about $-10°$C, also consistent with the basal ice temperature in other studies of glacial inception Marshall et al. (2002).

### A1 Finite Difference Approximation

We can use the flux form for the finite difference approximation. We use forward Euler and second order centered difference 25 as seen in Figure A3, where $n$ is the time step, $i$ is the node in the $x$ direction, and $j$ is the node in the $y$ direction.

$$\frac{H_{i,j}^{n+1} - H_{i,j}^n}{\Delta t} = \frac{D_{i+\frac{1}{2},j}^n \left( \frac{H_{i+1,j}^{*n} - H_{i,j}^{*n}}{\Delta x_i} \right) + D_{i-\frac{1}{2},j}^n \left( \frac{H_{i,j}^{*n} - H_{i-1,j}^{*n}}{\Delta x_{i-1}} \right)}{\frac{1}{2}(\Delta x_i + \Delta x_{i-1})} + \frac{D_{i,j+\frac{1}{2}}^n \left( \frac{H_{i,j+1}^{*n} - H_{i,j}^{*n}}{\Delta y_j} \right) + D_{i,j-\frac{1}{2}}^n \left( \frac{H_{i,j}^{*n} - H_{i,j-1}^{*n}}{\Delta y_{j-1}} \right)}{\frac{1}{2}(\Delta y_j + \Delta y_{j-1})} + G \tag{A3}$$

The numerical expression for the diffusivity is presented in Equation A7

$$D^n_{i+\frac{1}{2},j} = AH^{m+2}_{i+\frac{1}{2},j}\Big[\Big(\frac{H^*_{i+1,j} - H^*_{i,j}}{\Delta x_i}\Big)^2 + \Big(\frac{1}{2}\Big(\frac{H^*_{i+1,j+1} - H^*_{i+1,j-1}}{\Delta y_{j-1} + \Delta y_j} + \frac{H^*_{i,j+1} - H^*_{i,j-1}}{\Delta y_{j-1} + \Delta y_j}\Big)\Big)^2\Big]^{(m-1)/2} \tag{A4}$$

$$D^n_{i-\frac{1}{2},j} = AH^{m+2}_{i-\frac{1}{2},j}\Big[\Big(\frac{H^*_{i,j} - H^*_{i-1,j}}{\Delta x_{i-1}}\Big)^2 + \Big(\frac{1}{2}\Big(\frac{H^*_{i,j+1} - H^*_{i,j-1}}{\Delta y_{j-1} + \Delta y_j} + \frac{H^*_{i-1,j+1} - H^*_{i-1,j-1}}{\Delta y_{j-1} + \Delta y_j}\Big)\Big)^2\Big]^{(m-1)/2} \tag{A5}$$

$$D^n_{i,j+\frac{1}{2}} = AH^{m+2}_{i,j+\frac{1}{2}}\Big[\Big(\frac{1}{2}\Big(\frac{H^*_{i+1,j+1} - H^*_{i-1,j+1}}{\Delta x_{i-1} + \Delta x_i} + \frac{H^*_{i+1,j} - H^*_{i-1,j}}{\Delta x_{i-1} + \Delta x_i}\Big)\Big)^2 + \Big(\frac{H^*_{i,j+1} - H^*_{i,j}}{\Delta y_j}\Big)^2\Big]^{(m-1)/2} \tag{A6}$$

$$D^n_{i,j-\frac{1}{2}} = AH^{m+2}_{i,j-\frac{1}{2}}\Big[\Big(\frac{1}{2}\Big(\frac{H^*_{i+1,j+1} - H^*_{i-1,j+1}}{\Delta x_{i-1} + \Delta x_i} + \frac{H^*_{i+1,j} - H^*_{i-1,j}}{\Delta x_{i-1} + \Delta x_i}\Big)\Big)^2 + \Big(\frac{H^*_{i,j} - H^*_{i,j-1}}{\Delta y_{j-1}}\Big)^2\Big]^{(m-1)/2} \tag{A7}$$

Finally, our use of flux form requires Equation A11.

$$H^{m+2}_{i+\frac{1}{2},j} = \Big(\frac{H_{i+1,j} + H_{i,j}}{2}\Big)^{m+2} \tag{A8}$$

$$H^{m+2}_{i-\frac{1}{2},j} = \Big(\frac{H_{i,j} + H_{i-1,j}}{2}\Big)^{m+2} \tag{A9}$$

$$H^{m+2}_{i,j+\frac{1}{2}} = \Big(\frac{H_{i,j+1} + H_{i,j}}{2}\Big)^{m+2} \tag{A10}$$

$$H^{m+2}_{i,j-\frac{1}{2}} = \Big(\frac{H_{i,j} + H_{i,j-1}}{2}\Big)^{m+2} \tag{A11}$$

This model uses a time step of one year. At each time step, a new ice thickness is calculated based the equations above. Then with the new ice thickness and basal topography known, a new surface elevation is calculated ($H^* = B + H$).

The edge and center of the ice sheet can cause $D = 0$, which is unrealistic behavior. Thus, we set a minimum $D$, but as noted by Oerlemans (1981) and confirmed in our simulations, the ice model does not depend strongly on this choice.

We also characterize the mass balance according to elevation, where $G$ is actually $G(H^*)$. Thus, the mass balance is also
updated after each time step, since it depends on $H^*$. For our main set of simulations, we characterized the mass balance using elevation bins.

*Author contributions.*  LB performed the research work, all authors contributed to the planning, interpretation and writing of manuscript.

*Competing interests.*  The authors declare they have no conflict of interest.

*Acknowledgements.*  This work was funded by the Harvard Climate Change solutions fund, and by the NSF P2C2 program, grant OCE-
1602864. Timothy Cronin was supported by NSF grant AGS-1740533. Eli Tziperman thanks the Weizmann Institute for its hospitality during parts of this work. We would like to acknowledge high-performance computing support from Yellowstone (ark:/85065/d7wd3xhc) provided by NCAR's Computational and Information Systems Laboratory, sponsored by the National Science Foundation.

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
