# Peer review of "The Role of Regional Feedbacks in Glacial Inception on Baffin Island: The Interaction of Ice Flow and Meteorology"

_Climate of the Past, 2018_

## Referee Comment (RC1) · Anonymous Referee #1 · 5 Mar 2018

General comments:

This paper follows a long line of modeling studies on the last glacial inception ∼115,000 years before present. Using climate models with the 115 ka Earth orbital configuration (or 116 ka in some studies), sometimes coupled with ice flow models, there is a long-standing problem of not being able to simulate rapid ice-cap grown over Baffin Island and subsequent ice expansion over that region during the ensuing several thousand years, as indicated by the bulk of geologic evidence. These new results continue in the same vein, and find very little ice growth compared to the consensus "observed" view.

Notable features in this study are the use of a high-resolution regional climate model over Baffin Island (WRF, 20 km), asynchronous coupling with a dynamic ice cap model, elevation binning of surface mass balance, and negative feedback with anticyclonic

flow warming air at the ice margins. The introduction contains a helpful and reasonably thorough review of the long line of previous modelling work, and an outline of the observational basis. The paper is well organized and clear throughout. However, I have several major concerns with the methodology, listed below.

Specific comments:

(1) The RCM is forced at the lateral boundaries by ECMWF-reanalyzed meteorology for a modern year (1985-1986). External forcings related to 115 ka, i.e., Earth orbit and $CO_2$ level, are only applied in the RCM. The RCM's 100-km outer domain, shown in Figs. 8-11, covers much of North America and Greenland and nearby oceans, but not the entire Arctic or northern Eurasia. Consequently it is missing some of the large-scale forcing on hemispheric and semi-hemispheric scales at 115 ka, including variations in low-order planetary waves, due to the GCM boundary influence. More importantly, ocean surface temperatures and sea ice are prescribed in the RCM from the GCM (I think), and so remain at their modern state; in reality they would be strongly affected by the 115 ka orbital perturbations and influence Baffin Island climate. Also, within North America, the RCM physics contains no snow-masking albedo feedback due to vegetation ecotone shifts. All of these hemispheric-to-continental scale processes and feedbacks have been identified in previous modeling studies (see Introduction) as potentially significant players in cooling over Baffin Island and ice-cap initiation at 115 ka, but are muted or absent in the RCM simulations here.

To remedy this, I suggest that a GCM should be used, not modern reanalysis, with the GCM physics including ocean dynamics and sea ice, and with the GCM orbit changed to 115 ka. Preferably both the GCM and RCM would have vegetation feedbacks. Some of this is discussed on pg. 19, but should be implemented in my opinion.

(2) The paper presents results from a "WRF control simulation", described on pg. 5, line 27 and shown in subsequent figure panels. It is not entirely clear from the text, but I think this is really the first step in the asynchronous sequence, and uses 115 ka

orbit and reduced $CO_2$ (pg. 6, line 3). So all the differences from the second iteration in Figs. 4b, 5b, et seq. are due just to the initial ice cap growth in the first ice model integration. This "control" simulation is not a true modern simulation, with all-modern forcing (orbit and CO2). Such a run is described on pg. 5, lines 17-20, but not used again in the paper.

I suggest adding figures showing a basic sensitivity test, comparing that run (a true "modern control" with modern orbit and CO2) with the first WRF iteration run (the "WRF control" here, with 115 ka orbit, reduced CO2, still modern ice cap). And each driven by separate GCM simulations of modern and 115 ka climates, respectively, as suggested in point # 1 above. First, the modern RCM run should be checked to agree roughly with modern observed summer air temperatures, precipitation and surface mass balance (SMB) especially over Baffin Island (as it does according to pg. 5, lines 17-20).

Then an important figure should show differences in RCM summer air temperatures between the two runs, both for the whole outer domain (cf. Fig. 9b) and the inner domain (cf. Fig. 4b). The latter would immediately assess the viability of the whole scenario - i.e., qualitatively speaking, in order to produce major ice cap expansion, there needs to be at least a few degrees C of summertime cooling over the Baffin Island region, hopefully accompanied by some increase in annual snowfall. This basic cooling from truly modern conditions can then be contrasted later in the paper with the negative feedback presented here, where initial ice growth produces anticyclonic flow that warms the air around the ice margins.

(3) The use of just one modern year of ECMWF reanalysis does not adequately capture the mean (or interannual variability) of climate forcing. The choice of 1985-1986 as an extremely cold and wet year over Baffin Island bears an unknown relationship to the mean SMB forcing on century to millennial timescales that mainly determines ice growth. At a minimum, a GCM should be run for one (or two) decades, and the RCM run also through all those years, to give some idea of the mean SMB over Baffin Island. Choosing just one GCM year (or reanalysis, as here) can seriously skew the centuriesscale ice growth, due to the interannual variations of that single year.

(4) The resolution of the ice model (20 km, same as RCM), combined with the elevation-binning of the SMB calculations, may not be sufficient to capture the true overall mass balance and dynamic advance of the ice cap margins. The paper appropriately references van den Berg et al. (2006), who dramatically show that the ice grid needs to be fine enough to resolve the steeply sloping ice-cap surface in the ablation zone, over which SMB varies rapidly due mainly to the atmospheric lapse rate, from ∼zero at the equilibrium line to strongly negative at the ice edge. If the grid only has a few boxes within this zone, and there are large changes in surface ice elevation between neighboring boxes, then subtle changes in climate and the area-integrated SMB may not be captured accurately if at all. The degradation of results depends also on the amplitude of climate forcing, and the method of downscaling SMB to the ice model grid, and has probably occurred to varying degrees in previous inception studies.

van den Berg et al.'s test cases are ∼1000-km ice-sheet profiles, for which grid sizes of 5 km or less are needed for roughly accurate results (their Fig. 3). Here, the Baffin Island ice caps are much smaller, and the model's 20-km grid has only a few boxes within their narrow marginal ablation zones (see Fig. 1a, along SW-NE steepest-descent flow lines), which is probably not capturing true ice-cap advance. Judging from van den Berg et al.'s results, a much finer grid for the ice model should be used to ascertain the true behavior, on the order of a few to 1 km, at least until the initial ice caps grow much larger.

(5) Also, the elevation binning procedure may be contributing to the problem. Although not completely clear, I think the elevation binning (Fig. 1c) is done after each WRF integration, and the "bin line" (as in Fig. 3) is used to specify mass balance as a function of elevation for all points through the next ice model integration. However, the scatter in Fig. 1c shows that SMB is strongly influenced by factors other than elevation. In particular, SMB values around the edges of the ice cap, which are important in allowing or preventing ice advance, may be quite inaccurately represented by the procedure. An

alternative method would be to save mean monthly air temperatures and precipitation from the previous RCM integration, and downscale them to the surface elevation of all ice model grid points (by lateral interpolation, and vertical lapse-rate correction), and perform a calculation for annual SMB at each ice grid point, still including refreezing in a simplified way. This could also be used for "hypothetical" ice locations with negative SMB adjacent to the current edge, which are not available directly from WRF (pg. 6, line 5), into which ice can potentially expand.

Technical comments:

pg. 4, line 23: Perhaps basal topography (B) should be listed as an input to the ice model, not surface elevation (H*) or ice thickness (H) which are outputs. Unless H is meant as an initial condition(?).

pg. 22, line 16: For the calculation of T(z) in Appendix A, it is probably adequate to assume a linear conductive T(z) profile from bed to surface, as done here. But it could be augmented using the analytic "Robin" solution that accounts for vertical ice advection given the local SMB (e.g. Cuffey and Patterson, 2010, pg. 217-218, referenced here).

Once the basal ice temperatures are calculated, a check can be made that they are below freezing, and so are consistent with the assumption of zero sliding velocities in the ice model (pg. 4, line 8).

---

## Referee Comment (RC2) · Anonymous Referee #2 · 17 Apr 2018

General Comments:

This is an interesting paper concerning initiation of glaciation on Baffin Island by combining a regional atmospheric simulation with a straightforward ice flow model using a slightly modified contemporary year favorable to inception. The need for high spatial resolution is emphasized. My comments focus on the key aspect of this paper, namely the anticyclonic-cyclonic couplet that causes the warming that limits the ice growth. Is this real or an artifact of the WRF simulation? The reason I raise this question is that it is well known that regional models can develop anomalous circulations within their domains while matching conditions specified on the lateral boundary (e.g., Glisan et al., 2013: Effects of spectral nudging in WRF on Arctic temperature and precipitation simulations, J. Climate). Even if the couplet is not artificial a somewhat

different orientation/intensity could lead to different advection conditions, decreasing or even eliminating the warm air advection. So: 1. When you simulated the present-day climate was there any evidence of the above couplet compared to ERA-interim? 2. Rather than the differences in Figure 8, what do the full 500-hPa height fields look like for iterations 2 and 10? 3. Can you develop more compelling arguments for the reality of your circulation results?

More generally, the lateral boundary conditions for your model could be very different than what you specified due to climate system feedbacks as a result of reduced summer insolation so nesting a regional simulation in a GCM simulation for inception time might be the best next step in your modeling. Yet another rendition of the altered environment around 115k yr ago is Otieno et al. 2011: Atmospheric circulation anomalies due to 115k yr BP climate forcing are dominated by changes in the North Pacific Ocean. Clim. Dyn.

---

## Author Comment (AC1) · 15 May 2018

HARVARD UNIVERSITY
DEPARTMENT OF EARTH AND PLANETARY SCIENCES AND
SCHOOL OF ENGINEERING AND APPLIED SCIENCES
20 OXFORD ST, CAMBRIDGE, MASSACHUSETTS, 02138
lbirch@seas.harvard.edu
Leah Birch, Timothy Cronin and Eli Tziperman

May 15, 2018

Prof. Marit-Solveig Seidenkrantz
Handling Editor
Climate of the Past

Dear Prof. Seidenkrantz and CP Editors,

Please find enclosed the author response comments for "Glacial Inception on Baffin Island: The Interaction of Ice Flow and Meteorology", which we would like to submit for publication in Climate of the Past. We found the two reviewers' comments most helpful, and plan to address them in a revised manuscript as described in our response below. In particular, our goal in the revised manuscript is to clarify our decision to use a regional climate model rather than a global climate model and to demonstrate the robustness of the anomalous circulation that is part of the negative feedback we identify. We thank the reviewers for their comments and guidance, and we feel that the revised manuscript will be considerably improved as a result.

> Sincerely yours,
> Leah Birch, Timothy Cronin
> and Eli Tziperman

**Reviewer 1**

- This paper follows a long line of modeling studies on the last glacial inception ∼115,000 years before present. Using climate models with the 115 ka Earth orbital configuration (or 116 ka in some studies), sometimes coupled with ice flow models, there is a longstanding problem of not being able to simulate rapid ice-cap grown over Baffin Island and subsequent ice expansion over that region during the ensuing several thousand years, as indicated by the bulk of geologic evidence. These new results continue in the same vein, and find very little ice growth compared to the consensus "observed" view.
Notable features in this study are the use of a high-resolution regional climate model over Baffin Island (WRF, 20 km), asynchronous coupling with a dynamic ice cap model, elevation binning of surface mass balance, and negative feedback with anticyclonic flow warming air at the ice margins. The introduction contains a helpful and reasonably thorough review of the long line of previous modelling work, and an outline of the observational basis. The paper is well organized and clear throughout. However, I have several major concerns with the methodology, listed below

  Thank you for taking the time to read and for the thorough and constructive comments. In our revised manuscript, we will address your concerns and clarify the motivation for our model formulation and experiment design, in particular regarding the choices involved in the use of a regional climate model.

- Specific comments: (1) The RCM is forced at the lateral boundaries by ECMWF-reanalyzed meteorology for a modern year (1985-1986). External forcings related to 115 ka, i.e., Earth orbit and CO2 level, are only applied in the RCM. The RCMs 100-km outer domain, shown in Figs. 8-11, covers much of North America and Greenland and nearby oceans, but not the entire Arctic or northern Eurasia. Consequently it is missing some of the large-scale forcing on hemispheric and semi-hemispheric scales at 115 ka, including variations in low-order planetary waves, due to the GCM boundary influence.

  True, in an attempt to isolate the role of local processes over North America, we are neglecting circulation changes. This is consistent with Otieno et al. (2012) who used a coupled ocean atmosphere GCM, found a similar amount of cooling over Baffin Island during the summer (around 4C), and noted a lack of change in the Rossby waves. Wells (1983) also looked at Rossby waves during LGM compared to present day, and they found wave numbers of 4-5 are typical today, while at the height of the Laurentide a wave number is 3. This is not to say that there would definitely be no changes in planetary waves

at the beginning, but often in GCM simulations, only large ice sheets cause substantial deviations in the planetary waves. For instance, a large European ice sheet can cause cooling over the Laurentide (Beghin et al., 2014), but a large European Ice Sheet during inception may not be realistic (Bonelli et al., 2009; Ganopolski et al., 2010).

Yet, we agree that one expects changes to planetary waves due to the presence of ice sheets and topography changes. We hope to emphasize in our revised manuscript that, first, there is some debate regarding such changes, and second and more importantly, that our study neglects some feedbacks such as changes to the larger-scale circulation in order to focus on the role of those feedbacks that are included. We will clarify that our negative results may be due to these missing feedbacks, and that our study could therefore be complemented in the future by one in which the boundary conditions are modified accordingly, to test for feedbacks neglected here.

To emphasize our focus on local feedbacks, we will rename our manuscript to **"Role of Local Feedbacks in the Glacial Inception on Baffin Island: The Interaction of Ice Flow and Meteorology"**.

- More importantly, ocean surface temperatures and sea ice are prescribed in the RCM from the GCM (I think), and so remain at their modern state; in reality they would be strongly affected by the 115 ka orbital perturbations and influence Baffin Island climate.

This is an important comment, in the spirit of the previous one, and our response will be again divided into two parts. First, we will point out again in this context that our objective is to isolate local feedbacks and therefore we neglect some others. Second, we will explain that there is still debate on how much and how fast SSTs and sea ice would change during glacial inception. With present day fluxes on the boundaries we decided to keep SSTs at modern values for consistency, and investigate the "warm" interglacial temperature ocean scenario, which does have support in observations (Ruddiman et al., 1980). Cortijo et al. (1994) noted that the northern Atlantic remained close to modern day temperatures through 115 kya until glacial stage 5d around 110 kya. Vimeux et al. (1999) argued that observations support a warm ocean, similar to an interglacial, during inception, though their focus is on the souther hemisphere. Stokes (1955) and Gildor and Tziperman (2000, 2001) also argued that the oceans change slower than the atmosphere, so the oceans may need to remain warmer and ice free at the beginning of inception.

We will supplement our discussion of this issue using additional references, including studies that show that colder oceans may amplify the inception process

(Khodri et al., 2001), as discussed in the more modern analogue of the Little Ice Age by Lehner et al. (2013). On the other hand, Otieno et al. (2012) found few changes in temperature in the Atlantic, and Meissner and Gerdes (2002) noted with their ocean model that North Atlantic remained warmer. Jochum et al. (2012) also argued that with CCSM4 ocean feedbacks are not necessary for glacial inception, but also did not rule out the role of ocean cooling.

- Also, within North America, the RCM physics contains no snow-masking albedo feedback due to vegetation ecotone shifts. All of these hemispheric-to-continental scale processes and feedbacks have been identified in previous modeling studies (see Introduction) as potentially significant players in cooling over Baffin Island and ice-cap initiation at 115 ka, but are muted or absent in the RCM simulations here.

  We agree that vegetation changes are a potentially important part of the glacial inception process. Related studies often refer to the treeline moving southward (Goñi et al., 2005; Calov et al., 2005), which occurs later in the inception process. Baffin Island vegetation in this model is notably missing large trees (Fréchette et al., 2006), thus vegetation feedbacks may be less important on Baffin Island at the very beginning of the inception process. In any case, we will clarify this issue of vegetation vs. treeline and the relevant literature in our revised manuscript.

- To remedy this, I suggest that a GCM should be used, not modern reanalysis, with the GCM physics including ocean dynamics and sea ice, and with the GCM orbit changed to 115 ka. Preferably both the GCM and RCM would have vegetation feedbacks. Some of this is discussed on pg. 19, but should be implemented in my opinion.

  We agree that using boundary conditions from a glacial simulation of a GCM is a great next step to capture additional feedbacks, like planetary waves and the down wind effect of the Eurasian Ice sheet changes on topography. Additional feedbacks could be discovered with such simulations. Yet we do feel that there is significant value in exploring the role of local feedbacks, especially once we clarify our objectives in a revised manuscript, and emphasize our goal by renaming the manuscript as described above.

- (2) The paper presents results from a "WRF control simulation", described on pg. 5, line 27 and shown in subsequent figure panels. It is not entirely clear from the text, but I think this is really the first step in the asynchronous sequence, and uses 115 ka orbit and reduced CO2 (pg. 6, line 3). So all the differences from the second iteration in Figs. 4b, 5b, et seq. are due just to

the initial ice cap growth in the first ice model integration. This "control" simulation is not a true modern simulation, with all-modern forcing (orbit and CO2). Such a run is described on pg. 5, lines 17-20, but not used again in the paper. I suggest adding figures showing a basic sensitivity test, comparing that run (a true "modern control" with modern orbit and CO2) with the first WRF iteration run (the "WRF control" here, with 115 ka orbit, reduced CO2, still modern ice cap). And each driven by separate GCM simulations of modern and 115 ka climates, respectively, as suggested in point # 1 above. First, the modern RCM run should be checked to agree roughly with modern observed summer air temperatures, precipitation and surface mass balance (SMB) especially over Baffin Island (as it does according to pg. 5, lines 17-20). Then an important figure should show differences in RCM summer air temperatures between the two runs, both for the whole outer domain (cf. Fig. 9b) and the inner domain (cf. Fig. 4b). The latter would immediately assess the viability of the whole scenario - i.e., qualitatively speaking, in order to produce major ice cap expansion, there needs to be at least a few degrees C of summertime cooling over the Baffin Island region, hopefully accompanied by some increase in annual snowfall. This basic cooling from truly modern conditions can then be contrasted later in the paper with the negative feedback presented here, where initial ice growth produces anticyclonic flow that warms the air around the ice margins.

Thank you for this comment. We realize now that our terminology may have been confusing and we will attempt to emphasize that our first 115kya simulation is Iteration 1.

As for an actual modern control simulation, we will make sure in our revision to refer the readers to our previous paper (Birch et al., 2017), which illustrates the differences in 115 kya and present day insolation. The resolution was higher, at 4km, but we found that using 20 km resolution did not significantly alter those results. The highest peaks of the Penny Ice Cap was colder with 4km, but at the high altitudes this does not differ the mass balance, as melting is already not occurring there. To address this comment, we will also include in the revised manuscript a plot of the temperature differences between model results using the present day insolation and 115 kya insolation in both the 100km outer domain and 20 km inner domains.

- (3) The use of just one modern year of ECMWF reanalysis does not adequately capture the mean (or interannual variability) of climate forcing. The choice of 1985-1986 as an extremely cold and wet year over Baffin Island bears an unknown relationship to the mean SMB forcing on century to millennial

timescales that mainly determines ice growth. At a minimum, a GCM should be run for one (or two) decades, and the RCM run also through all those years, to give some idea of the mean SMB over Baffin Island. Choosing just one GCM year (or reanalysis, as here) can seriously skew the centuries-scale ice growth, due to the interannual variations of that single year.

It is true, of course, that a longer averaging of the forcing, as obtained from a climate model, would have advantages. However, given the persistent difficulties in simulating glacial inception using such GCMs, our goal here was to minimize the introduction of biases from global climate simulations, and stick to observations (reanalysis) and to local feedbacks as closely as possible. That has a price, as clearly pointed out by the reviewer, but we feel our approach is at least self-consistent and transparent. We hope to further clarify and elaborate on the motivation for this experiment design in our revision.

- (4) The resolution of the ice model (20 km, same as RCM), combined with the elevation binning of the SMB calculations, may not be sufficient to capture the true overall mass balance and dynamic advance of the ice cap margins. The paper appropriately references van den Berg et al. (2006), who dramatically show that the ice grid needs to be fine enough to resolve the steeply sloping ice-cap surface in the ablation zone, over which SMB varies rapidly due mainly to the atmospheric lapse rate, from ∼zero at the equilibrium line to strongly negative at the ice edge. If the grid only has a few boxes within this zone, and there are large changes in surface ice elevation between neighboring boxes, then subtle changes in climate and the area-integrated SMB may not be captured accurately if at all. The degradation of results depends also on the amplitude of climate forcing, and the method of downscaling SMB to the ice model grid, and has probably occurred to varying degrees in previous inception studies. van den Berg et al.'s test cases are ∼1000-km ice-sheet profiles, for which grid sizes of 5 km or less are needed for roughly accurate results (their Fig. 3). Here, the Baffin Island ice caps are much smaller, and the model's 20-km grid has only a few boxes within their narrow marginal ablation zones (see Fig. 1a, along SW-NE steepest-descent flow lines), which is probably not capturing true ice-cap advance. Judging from van den Berg et al.'s results, a much finer grid for the ice model should be used to ascertain the true behavior, on the order of a few to 1 km, at least until the initial ice caps grow much larger.

We agree that a higher resolution of the ice model would be optimal and did run the ice model at a 4 km resolution as well (Birch, PhD thesis, 2017). We found the same results, as seen in Figure 1 below. In our revision, we will include a figure of these results and further discuss the issue.

[Figure]

(a) 20km Resolution  (b) 4km Resolution

Figure 1: Ice extent after 2nd Iteration for 20km and 4m Resolutions, using the mass balance from 20km WRF Simulation

- (5) Also, the elevation binning procedure may be contributing to the problem. Although not completely clear, I think the elevation binning (Fig. 1c) is done after each WRF integration, and the "bin line" (as in Fig. 3) is used to specify mass balance as a function of elevation for all points through the next ice model integration. However, the scatter in Fig. 1c shows that SMB is strongly influenced by factors other than elevation. In particular, SMB values around the edges of the ice cap, which are important in allowing or preventing ice advance, may be quite inaccurately represented by the procedure. An alternative method would be to save mean monthly air temperatures and precipitation from the previous RCM integration, and downscale them to the surface elevation of all ice model grid points (by lateral interpolation, and vertical lapse-rate correction), and perform a calculation for annual SMB at each ice grid point, still including refreezing in a simplified way. This could also be used for "hypothetical" ice locations with negative SMB adjacent to the current edge, which are not available directly from WRF (pg. 6, line 5), into which ice can potentially expand.

    Our binning procedure and the resulting SMB forcing recipe, while simple, are consistent with the way most ice sheet models are forced. More importantly, the results of this procedure, showing that ice elevation causes a negative warming feedback, should be robust regardless of how the ice elevation was calculated, as it is a result of the atmospheric model itself. We will further discuss and clarify these issues in our revision. We will also mention the related result that

finer bin sizes did not increase the expansion of ice.

- Technical comments:

  pg. 4, line 23: Perhaps basal topography (B) should be listed as an input to the ice model, not surface elevation (H*) or ice thickness (H) which are outputs. Unless H is meant as an initial condition(?).

  $H$ is the present day ice thickness the Ice Bridge Project, while the surface elevation $(H^*)$ is specified as in put in WRF. The basal topography is thus $B = H^* - H$. All are needed as inputs or initial conditions in the ice model and we will make this clearer in the revised Methods Section.

- pg. 22, line 16: For the calculation of $T(z)$ in Appendix A, it is probably adequate to assume a linear conductive $T(z)$ profile from bed to surface, as done here. But it could be augmented using the analytic "Robin" solution that accounts for vertical ice advection given the local SMB (e.g. Cuffey and Patterson, 2010, pg. 217-218, referenced here). Once the basal ice temperatures are calculated, a check can be made that they are below freezing, and so are consistent with the assumption of zero sliding velocities in the ice model (pg. 4, line 8).

  Thank you for this idea. We believe the Robin solution involves the figure on page 411 and the associated equations. We can discuss this additional check in our revised manuscript appendix, but the temperature at the base calculated from this set of equations is still below freezing at $-7°$C.

**Reviewer 2**

- This is an interesting paper concerning initiation of glaciation on Baffin Island by combining a regional atmospheric simulation with a straightforward ice flow model using a slightly modified contemporary year favorable to inception. The need for high spatial resolution is emphasized.
Thank you for the most helpful comments; we plan to follow up on your suggestion and make our argument for the anomalous anticyclonic-cyclonic circulation and the negative feedback stronger.

- My comments focus on the key aspect of this paper, namely the anticyclonic-cyclonic couplet that causes the warming that limits the ice growth. Is this real or an artifact of the WRF simulation? The reason I raise this question is that it is well known that regional models can develop anomalous circulations within their domains while matching conditions specified on the lateral boundary (e.g., Glisan et al., 2013: Effects of spectral nudging in WRF on Arctic temperature and precipitation simulations, J. Climate). Even if the couplet is not artificial a somewhat different orientation/intensity could lead to different advection conditions, decreasing or even eliminating the warm air advection.
Thank you for reminding us of this paper, which we will further discuss in our revised manuscript. We are not employing spectral (internal) nudging, and restrict nudging to the domain boundaries. The goal of spectral nudging is to prevent departures from the GCM used for boundary conditions, while for our objectives, such deviations are of interest and are therefore not constrained. When running a present day simulation, we find that the circulation here is similar to that found in ERA-Interim. The simulation with 115 kya insolation also has circulation similar in magnitude and direction. The differences come in once topography changes are introduced. This indicates that the anticyclonic-cyclonic response is likely not an artifact, and we will further explore and discuss this in the revised manuscript.

- So: 1. When you simulated the present-day climate was there any evidence of the above couplet compared to ERA-interim?
We did find that June was warmer by $\sim 1$ degree in WRF than ERA-iterim for the present day simulation, but we do not believe it is a pattern inherent in WRF. The circulation in WRF and ERA-Interim are similar in direction and magnitude, and the 1st iteration simulation with 115 kya insolation does not cause the couplet to appear. Our simulations with and without ice-topography changes, robustly indicate that the anomaly appears only once the ice topography on Baffin Island changes. We will show the circulation patterns in presentday simulations in order to address this issue in our revised manuscript.

- 2. Rather than the differences in Figure 8, what do the full 500-hPa height fields look like for iterations 2 and 10? 3. Can you develop more compelling arguments for the reality of your circulation results?

  The full 500 mbar height fields show the same flow pattern, and we will show these in a supplement added to our revised manuscript. We will also make the case more compellingly that these circulation anomalies do not show up unless the ice topography is changed. The anomalous response reveals that winds are not as strong from the north, which causes the warmer temperatures. We will also include further analysis of our simulations with topography or ice changes alone, particularly the geopotential height, instead of just the temperatures presented in our first submission.

- More generally, the lateral boundary conditions for your model could be very different than what you specified due to climate system feedbacks as a result of reduced summer insolation so nesting a regional simulation in a GCM simulation for inception time might be the best next step in your modeling.

  We agree that one expects changes to horizontal boundary conditions in a glacial world, due to planetary wave response to the developing ice sheets and other factors. We will emphasize in our revised manuscript that, first, there is some debate regarding such changes, and second and more importantly, that our study neglects some feedbacks such as changes to the larger-scale circulation in order to focus on the role of those feedbacks that are included. We will clarify that our negative results may be due to these missing feedbacks.

  To emphasize our focus on local feedbacks, we will rename our manuscript to **"Role of Local Feedbacks in the Glacial Inception on Baffin Island: The Interaction of Ice Flow and Meteorology".**

- Yet another rendition of the altered environment around 115k yr ago is Otieno et al. 2011: Atmospheric circulation anomalies due to 115k yr BP climate forcing are dominated by changes in the North Pacific Ocean. Clim. Dyn.

  Thank you for bringing the Otieno et al. (2012) paper to our attention. We have found it very useful, and we believe it emphasizes that circulation over the Atlantic may not have changed much by the time of inception. They found that the Pacific Ocean has the largest affect over the western part of North America, and causes substantial cooling there. The general circulation could be quite different, but the anti-cyclone is still there as noted by Herrington and Poulsen (2011) and Gregory et al. (2012). Otieno et al. (2012) also noted the formation of an anti-cyclone over Baffin Island.

**References**

[revised manuscript text omitted]

---

## Author Response (AR1)

HARVARD UNIVERSITY
DEPARTMENT OF EARTH AND PLANETARY SCIENCES AND
SCHOOL OF ENGINEERING AND APPLIED SCIENCES
20 OXFORD ST, CAMBRIDGE, MASSACHUSETTS, 02138
lbirch@seas.harvard.edu
Leah Birch, Timothy Cronin and Eli Tziperman

June 27, 2018

Prof. Marit-Solveig Seidenkrantz
Handling Editor
Climate of the Past

Dear Prof. Seidenkrantz and CP Editors,

Please find enclosed the revised manuscript and author response comments for "Glacial Inception on Baffin Island: The Interaction of Ice Flow and Meteorology", renamed following the reviewer comments to "The Role of Regional Feedbacks in Glacial Inception on Baffin Island: The Interaction of Ice Flow and Meteorology". The tracked changes appear at the end of this document and page number references refer to the revised manuscript.

We would like to thank you for the time to make our arguments clearer, and we hope to have emphasized our choice of model and location of glacial inception on Baffin Island. We found the two reviewers' comments most helpful and constructive, fully addressing them with figures, analyses and discussion. In response to the suggestions of reviewer 1, we now explain that the goal of this work is to study local feedbacks over Baffin Island that may lead to inception there, clarifying our decision to use a regional climate model. Based on comments by reviewer 2, we now explicitly discuss and demonstrate the robustness of the anomalous circulation and the negative feedback. We thank the reviewers for their comments and guidance, and we feel that the revised manuscript is considerably improved.

Sincerely yours,
Leah Birch, Timothy Cronin
and Eli Tziperman

**Reviewer 1**

- This paper follows a long line of modeling studies on the last glacial inception ∼115,000 years before present. Using climate models with the 115 ka Earth orbital configuration (or 116 ka in some studies), sometimes coupled with ice flow models, there is a longstanding problem of not being able to simulate rapid ice-cap grown over Baffin Island and subsequent ice expansion over that region during the ensuing several thousand years, as indicated by the bulk of geologic evidence. These new results continue in the same vein, and find very little ice growth compared to the consensus "observed" view.
  Notable features in this study are the use of a high-resolution regional climate model over Baffin Island (WRF, 20 km), asynchronous coupling with a dynamic ice cap model, elevation binning of surface mass balance, and negative feedback with anticyclonic flow warming air at the ice margins. The introduction contains a helpful and reasonably thorough review of the long line of previous modelling work, and an outline of the observational basis. The paper is well organized and clear throughout. However, I have several major concerns with the methodology, listed below

  Thank you for taking the time to read and for the thorough and constructive comments. In our revised manuscript, we address your concerns and clarify the motivation for our model formulation and experiment design, in particular regarding the choices involved in the use of a regional climate model.

- Specific comments: (1) The RCM is forced at the lateral boundaries by ECMWF-reanalyzed meteorology for a modern year (1985-1986). External forcings related to 115 ka, i.e., Earth orbit and CO2 level, are only applied in the RCM. The RCM's 100-km outer domain, shown in Figs. 8-11, covers much of North America and Greenland and nearby oceans, but not the entire Arctic or northern Eurasia. Consequently it is missing some of the large-scale forcing on hemispheric and semi-hemispheric scales at 115 ka, including variations in low-order planetary waves, due to the GCM boundary influence.

  We emphasize in the revised manuscript that, first, there is some debate regarding such changes (Page 3, Lines 22-24 and Page 22, Lines 20-22),

  > By prescribing boundary conditions from the modern climate, we do not consider any role of non-local ice sheet growth in modifying the planetary wave pattern, which could aid or hinder inception on Baffin Island. This is consistent both with other studies (Bonelli et al., 2009; Ganopolski et al., 2010) and with our intention to study the effect of regional feedbacks.

. . .
> We are assuming that the planetary waves response to the growing Laurentide ice sheet (Wells, 1983) was not significant until the size of the Laurentide was substantial, as has been found in other modeling studies of glacial inception (Otieno et al., 2012).

Second and more importantly, we note that that our study neglects some feedbacks such as changes to the planetary-scale circulation in order to focus on the role of those feedbacks that are included (Page 22, Lines 23-27).

> The strength of this negative regional feedback that prevents inception from proceeding, suggests that a reorganization of the planetary-scale Northern Hemisphere circulation could occur as part of the inception process and counter the regional negative feedback. We cannot capture such a change in the general circulation with our regional model and prescribed boundary conditions, indicating a study of non-local processes could complement this study of local Baffin Island feedbacks.

To emphasize our focus on local feedbacks, we have renamed our manuscript to **"Role of Regional Feedbacks in Glacial Inception on Baffin Island: The Interaction of Ice Flow and Meteorology"**.

- More importantly, ocean surface temperatures and sea ice are prescribed in the RCM from the GCM (I think), and so remain at their modern state; in reality they would be strongly affected by the 115 ka orbital perturbations and influence Baffin Island climate.

  This is an important comment, and our response is again divided into two parts. First, we point out in this context that our objective is to isolate local feedbacks and, second, that there is still debate on how much and how fast SSTs and sea ice would change during glacial inception. There is a detailed discussion on the downside and support to this approach in the revised discussion section, (Page 21, line 25 to Page 22, line 9), the beginning and end of which read as follows,

  > Allowing the SST to cool has been shown to induce cooling over land, even if the atmosphere is relatively warm (Yoshimori et al., 2002), while Kageyama and Valdes (2000) noted that storm tracks change with colder SSTs in a manner conducive for inception. In contrast, Stokes (1955) and Gildor and Tziperman (2000, 2001) argued that the oceans change slower than the atmosphere, so the oceans may remain warmer and ice free at the beginning of inception.

. . .

> With present day fluxes prescribed at the boundaries, and the focus
> of this study being local feedbacks, we keep SSTs at modern values
> for consistency, and effectively investigate the "warm" interglacial
> temperature ocean scenario, which does have support in observations
> (Ruddiman et al., 1980).

- Also, within North America, the RCM physics contains no snow-masking albedo feedback due to vegetation ecotone shifts. All of these hemispheric-to-continental scale processes and feedbacks have been identified in previous modeling studies (see Introduction) as potentially significant players in cooling over Baffin Island and ice-cap initiation at 115 ka, but are muted or absent in the RCM simulations here.

  This is another important point, but it may be more important later in the inception process. We now explicitly justify this assumption (Page 22, lines 17-18),

  > The southward treeline trend, though, has been shown to occur later
  > in the inception process (Goñi et al., 2005), justifying neglecting this
  > feedback.

- To remedy this, I suggest that a GCM should be used, not modern reanalysis, with the GCM physics including ocean dynamics and sea ice, and with the GCM orbit changed to 115 ka. Preferably both the GCM and RCM would have vegetation feedbacks. Some of this is discussed on pg. 19, but should be implemented in my opinion.

  We agree that using boundary conditions from a glacial simulation of a GCM is a great next step to capture additional feedbacks, like planetary waves and the down wind effect of the Eurasian Ice sheet changes on topography. Yet this would be a different study than this paper describes, and we do feel that there is significant value in exploring the role of local feedbacks, like regional ice-albedo and height-mass balance feedbacks. Such local feedbacks are expected to dominate some of the planetary-scale feedbacks at the very beginning of the inception process, as we make clearer now. Hopefully our clarification of the work objectives in the revised manuscript, and the renaming of the manuscript, make this point clearer than in the original manuscript.

- (2) The paper presents results from a "WRF control simulation", described on pg. 5, line 27 and shown in subsequent figure panels. It is not entirely clear from the text, but I think this is really the first step in the asynchronous sequence, and uses 115 ka orbit and reduced CO2 (pg. 6, line 3). So all the

differences from the second iteration in Figs. 4b, 5b, et seq. are due just to the initial ice cap growth in the first ice model integration.

Thank you for this comment. We realize now that our terminology may have been confusing and now consistently refer to our first 115kya simulation as Iteration 1.

- This "control" simulation is not a true modern simulation, with all-modern forcing (orbit and CO2). Such a run is described on pg. 5, lines 17-20, but not used again in the paper. I suggest adding figures showing a basic sensitivity test, comparing that run (a true "modern control" with modern orbit and CO2) with the first WRF iteration run (the "WRF control" here, with 115 ka orbit, reduced CO2, still modern ice cap).

We now show the results of the modern control simulations, as suggested (also shown previously in Birch et al. (2017) as Supplement Fig. 3.

- And each driven by separate GCM simulations of modern and 115 ka climates, respectively, as suggested in point # 1 above.

We believe that this would be an excellent next step to understand how large-scale changes may impact the inception area and interact with regional feedback found here, but feel this would be outside the scope of this study. We now discuss this throughout the manuscript, including in the revised conclusions (Page 24, lines 12-14),

> . . . Going a step further to more non-local feedbacks (such as SST changes) influenced by changing orbital parameters, a next step would be simulating the last glacial inception with a GCM and using the results as boundary conditions in a regional model.

- First, the modern RCM run should be checked to agree roughly with modern observed summer air temperatures, precipitation and surface mass balance (SMB) especially over Baffin Island (as it does according to pg. 5, lines 17-20). Then an important figure should show differences in RCM summer air temperatures between the two runs, both for the whole outer domain (cf. Fig. 9b) and the inner domain (cf. Fig. 4b). The latter would immediately assess the viability of the whole scenario - i.e., qualitatively speaking, in order to produce major ice cap expansion, there needs to be at least a few degrees C of summertime cooling over the Baffin Island region, hopefully accompanied by some increase in annual snowfall. This basic cooling from truly modern conditions

can then be contrasted later in the paper with the negative feedback presented here, where initial ice growth produces anticyclonic flow that warms the air around the ice margins.

This comment raises some very important points, and we now added a figure to the supplement to address them. For the modern RCM and explain (Page 6, Lines 7-23),

> With the above WRF configuration, we find a reasonable reproduction of the Arctic atmospheric state when we look at the full fields (Supplementary Figure 1). While there are deviations in geopotential and temperature between WRF and ECMWF, the differences in geopotential have been seen in other Arctic studies (Glisan and Gutowski, 2014). The temperature bias occurs from differences in the snow cover between the models. For instance, ECMWF has been noted to have too large of a snow-covered area on coasts (Drusch et al., 2004), promoting cooling on Baffin Island which is mostly coast at the reanalysis resolution. Also June is the cause of the large temperature bias because ECMWF still has snow on the ground, associated with delayed snow melt (Dutra et al., 2010), while the snow in WRF melts earlier. July and August only differ in temperature by about 1 degree, when the snow cover in ECMWF and WRF are more similar. With the warm temperature bias, WRF will likely not cause an artificial inception, which making it more suitable than models with a cold bias. Furthermore, ECMWF is known to have a cold bias (Bromwich et al., 2002b; Screen and Simmonds, 2011) and a significant low geopotential height in the central Arctic, of the same magnitude as the deviation we see in Supplementary Figure 1. Therefore, with WRF having the opposite biases, the Arctic climate simulated is not abnormal. Our WRF simulation matches observations of precipitation on the coast of Baffin Island, important for the surface mass balance, of about 300 mm per year (Zdanowicz et al., 2012). In any case, the warm bias in our WRF control run does not prevent the circulation from bringing relatively cold and wet conditions to Baffin Island (Bromwich et al., 2002a). Previous works also proceeded to look at climate sensitivity by comparing model experiments in spite of biases in the control run Porter et al. (2012), as we do here, and we therefore do not believe the biases would have a significant impact on the sensitivity results here.

Next, we added a figure showing the viability of our approach, as suggested, (page 6, line 28-31)

> Supplementary Fig. 4 shows cumulative mass-balance for present-day average conditions, present-day cold meteorology conditions, and cold meteorology plus 115 kya orbital forcing. The combination of moderately cool temperatures and reduced summer insolation leads to net snow accumulation, hinting that inception might be viable given sufficently positive regional feedbacks.

- (3) The use of just one modern year of ECMWF reanalysis does not adequately capture the mean (or interannual variability) of climate forcing. The choice of 1985-1986 as an extremely cold and wet year over Baffin Island bears an unknown relationship to the mean SMB forcing on century to millennial timescales that mainly determines ice growth. At a minimum, a GCM should be run for one (or two) decades, and the RCM run also through all those years, to give some idea of the mean SMB over Baffin Island. Choosing just one GCM year (or reanalysis, as here) can seriously skew the centuries-scale ice growth, due to the interannual variations of that single year.

We now explain explicitly mention this caveat (Page 6, line 31 to Page 7, line 2).

> A longer averaging of the forcing, as obtained from a climate model, would have some advantages. However, given the persistent difficulties in simulating glacial inception using such GCMs, our goal here is to minimize the introduction of biases from global climate simulations. We therefore prefer to choose the necessary cold and wet meteorology from the observed modern climate (reanalysis). We cannot explain how these ideal circulation patterns occur during inception, but a future study with a GCM might carefully explore this. We do note that forcing with a single-year meteorology may bias the ice growth patterns if that year is not representative of longer-term cold conditions.

- (4) The resolution of the ice model (20 km, same as RCM), combined with the elevation binning of the SMB calculations, may not be sufficient to capture the true overall mass balance and dynamic advance of the ice cap margins. The paper appropriately references van den Berg et al. (2006), who dramatically show that the ice grid needs to be fine enough to resolve the steeply sloping ice-cap surface in the ablation zone, over which SMB varies rapidly due mainly

to the atmospheric lapse rate, from ∼zero at the equilibrium line to strongly negative at the ice edge. If the grid only has a few boxes within this zone, and there are large changes in surface ice elevation between neighboring boxes, then subtle changes in climate and the area-integrated SMB may not be captured accurately if at all. The degradation of results depends also on the amplitude of climate forcing, and the method of downscaling SMB to the ice model grid, and has probably occurred to varying degrees in previous inception studies. van den Berg et al.'s test cases are ∼1000-km ice-sheet profiles, for which grid sizes of 5 km or less are needed for roughly accurate results (their Fig. 3). Here, the Baffin Island ice caps are much smaller, and the model's 20-km grid has only a few boxes within their narrow marginal ablation zones (see Fig. 1a, along SW-NE steepest-descent flow lines), which is probably not capturing true ice-cap advance. Judging from van den Berg et al.'s results, a much finer grid for the ice model should be used to ascertain the true behavior, on the order of a few to 1 km, at least until the initial ice caps grow much larger.

We now include a figure of the ice model run at a 4 km resolution (Birch, PhD thesis, 2017) in the supplementary information. The figure shows much more detailed structure, yet the same overall ice expansion as captured by the coarser model used in the paper, and discuss this as follows (Page 23, lines 20-22),

> . . . we find that simulations with a 4 km resolution (Supplementary Fig. 6) yield similar ice flow patterns to the 20 km results we present here.

- (5) Also, the elevation binning procedure may be contributing to the problem. Although not completely clear, I think the elevation binning (Fig. 1c) is done after each WRF integration, and the "bin line" (as in Fig. 3) is used to specify mass balance as a function of elevation for all points through the next ice model integration. However, the scatter in Fig. 1c shows that SMB is strongly influenced by factors other than elevation. In particular, SMB values around the edges of the ice cap, which are important in allowing or preventing ice advance, may be quite inaccurately represented by the procedure. An alternative method would be to save mean monthly air temperatures and precipitation from the previous RCM integration, and downscale them to the surface elevation of all ice model grid points (by lateral interpolation, and vertical lapse-rate correction), and perform a calculation for annual SMB at each ice grid point, still including refreezing in a simplified way. This could also be used for "hypothetical" ice locations with negative SMB adjacent to the current edge, which

are not available directly from WRF (pg. 6, line 5), into which ice can potentially expand.

The reviewer's understanding of our procedure is correct, which attempted to make clearer in the revised manuscript methods, page 7. We did not carry the suggested alternative procedure out as it is not obvious that it would be better, and that would require running all iterations of the coupled ice-sheet WRF model again. We discuss our procedure and present the alternative suggested in our revised manuscript as follows (discussion section, Page 23, lines 28-33),

> Our binning procedure and the resulting SMB forcing recipe, while simple, are consistent with the way many simple ice sheet models are forced. More importantly, the results of this procedure, showing that ice elevation causes a negative warming feedback, should be robust regardless of how the ice elevation was calculated, as it is a result of the atmospheric model itself. We find that finer bin sizes did not significantly affect the expansion of ice. However, it would be good to test alternative SMB recipes, such as using mean monthly air temperatures and precipitation from the previous WRF integrations, and downscale them to the surface elevation of ice grid points.

- Technical comments:
  pg. 4, line 23: Perhaps basal topography (B) should be listed as an input to the ice model, not surface elevation (H*) or ice thickness (H) which are outputs. Unless H is meant as an initial condition(?).

  These are inputs or initial conditions in the ice model, as we now clarify in the revised Methods Section (page 5, lines 6-7),

  > The ice model requires the specification of initial surface elevation $(H^*)$, initial ice thickness $(H)$, basal topography $(B)$ that is time-independent, and mass balance $(G)$ as an input for each time step.

- pg. 22, line 16: For the calculation of $T(z)$ in Appendix A, it is probably adequate to assume a linear conductive $T(z)$ profile from bed to surface, as done here. But it could be augmented using the analytic "Robin" solution that accounts for vertical ice advection given the local SMB (e.g. Cuffey and Patterson, 2010, pg. 217-218, referenced here). Once the basal ice temperatures are calculated, a check can be made that they are below freezing, and so are consistent with the assumption of zero sliding velocities in the ice model (pg. 4, line 8).

We calculated Robin's solution as suggested (we find this in Cuffey and Patterson, 2010, pp 410-411, equation 9.16), show it in Figure 1 shown in this response letter, and discuss it in the revised manuscript as follows (Page 25, lines 20-21),

> We find that using Robin's solution (Cuffey and Paterson, 2010) instead of a linear profile confirms that the basal temperature is below freezing, consistent with our shallow ice model formulation. The basal temperature we use is not far from about $-10°C$, also consistent with the basal ice temperature in other studies of glacial inception Marshall et al. (2002).

[Figure]

Figure 1: Basal ice temperature using a linear solution and using Robin's solution.

**Reviewer 2**

- This is an interesting paper concerning initiation of glaciation on Baffin Island by combining a regional atmospheric simulation with a straightforward ice flow model using a slightly modified contemporary year favorable to inception. The need for high spatial resolution is emphasized.
  Thank you for the most helpful comments; we followed up on your suggestions and hopefully made our arguments for the anomalous anticyclonic-cyclonic circulation and the negative feedback stronger as further discussed below.

- My comments focus on the key aspect of this paper, namely the anticyclonic-cyclonic couplet that causes the warming that limits the ice growth. Is this real or an artifact of the WRF simulation? The reason I raise this question is that it is well known that regional models can develop anomalous circulations within their domains while matching conditions specified on the lateral boundary (e.g., Glisan et al., 2013: Effects of spectral nudging in WRF on Arctic temperature and precipitation simulations, J. Climate). Even if the couplet is not artificial a somewhat different orientation/ intensity could lead to different advection conditions, decreasing or even eliminating the warm air advection.
  We do agree that this is an important issue. We show evidence that the warming feedback we describe is not artificial, yet we added caveats to this as a potential issue that may affect our results or similar future studies. Specifically, we first explain (Methods section, page 5, lines 25-31),

  > Lateral boundary conditions are prescribed at the edges of the outer domain, yet no nudging to observations is used in the domain interior. Such spectral nudging is often used in regional models to eliminate artificial circulation patterns that develop in spite of the prescribed boundary conditions (Glisan et al., 2013). Comparing our control run to ECMWF, we find (supplementary Fig. 1) non-negligible deviations in geopotential height and temperature. However, the anomaly over Baffin Island is opposite that of our anti-cyclone response. Yet we show below that the anomalous circulation that develops in response to ice growth in our simulations is likely unrelated to these anomalies, as they develop gradually in response to the growing ice height.

  Please see more in response to the following question.

- So: 1. When you simulated the present-day climate was there any evidence of the above couplet compared to ERA-interim?

We are now presenting comparisons with ECMWF in the supplement, we would argue that same anomaly over Baffin Island does not occur (Page 6, lines 7-23), and we argue that the WRF response occurs after topography changes are induced.

With the above WRF configuration, we find a reasonable reproduction of the Arctic atmospheric state when we look at the full fields (Supplementary Figure 1). While there are deviations in geopotential and temperature between WRF and ECMWF, the differences in geopotential have been seen in other Arctic studies (Glisan and Gutowski, 2014). The temperature bias occurs from differences in the snow cover between the models. For instance, ECMWF has been noted to have too large of a snow-covered area on coasts (Drusch et al., 2004), promoting cooling on Baffin Island which is mostly coast at the reanalysis resolution. Also June is the cause of the large temperature bias because ECMWF still has snow on the ground, associated with delayed snow melt (Dutra et al., 2010), while the snow in WRF melts earlier. July and August only differ in temperature by about 1 degree, when the snow cover in ECMWF and WRF are more similar. With the warm temperature bias, WRF will likely not cause an artificial inception, which making it more suitable than models with a cold bias. Furthermore, ECMWF is known to have a cold bias (Bromwich et al., 2002b; Screen and Simmonds, 2011) and a significant low geopotential height in the central Arctic, of the same magnitude as the deviation we see in Supplementary Figure 1. Therefore, with WRF having the opposite biases, the Arctic climate simulated is not abnormal. Our WRF simulation matches observations of precipitation on the coast of Baffin Island, important for the surface mass balance, of about 300 mm per year (Zdanowicz et al., 2012). In any case, the warm bias in our WRF control run does not prevent the circulation from bringing relatively cold and wet conditions to Baffin Island (Bromwich et al., 2002a). Previous works also proceeded to look at climate sensitivity by comparing model experiments in spite of biases in the control run Porter et al. (2012), as we do here, and we therefore do not believe the biases would have a significant impact on the sensitivity results here.

- 2. Rather than the differences in Figure 8, what do the full 500-hPa height fields look like for iterations 2 and 10? 3. Can you develop more compelling

First, we added supplementary Fig. 2 with the full 500-hPa height fields for iterations 1,2, and 10. The geopotential height anomalies for the No H and H Only simulations are also presented alongside the iteration 2 anomaly.
Then as for a more compelling argument, we added to the paper itself the new Fig. 10 with two scatter plots, and discuss them as follows (Page 13, line 33 to Page 15, line 15),

> It is known that regional models can develop artificial anomalous circulations (Glisan et al., 2013) and it is important to verify that the negative feedback identified here is not strongly affected by such artifacts. Fig. 10 shows two scatter plots of the temperature anomaly and the geopotential anomaly as function of ice height for iterations 2-10. Clearly the anomalous circulation and heating develops gradually with the ice growth, suggesting that the anomalous circulation is driven by the changing ice topography and is likely not an artifact. We further investigate the effect of topography in the following section. The geopotential anomaly increases with ice height, with the exception of the last iteration when the topography undergoes a new large area of ice appears. Given that the temperature anomaly is still strong in this last iteration, the feedback is still valid through this last iteration. The response we find is consistent with previous studies who noted that the size of the mountain is critical to the orographic response (Chen and Lin, 2005; Cook and Held, 1988) and slight changes to topography can cause significant changes (Roe and Lindzen, 2001). We speculate that the physical mechanism underlying the topography-induced anticyclone might be related to mixing of shallow-water potential vorticity (PV) by eddies near a localized high point in a weak background flow. Lateral eddy diffusion near a high point would stir high-PV air (due to lower thicknesses) away from the high terrain. The anti-cyclone is created by the influx of lower-PV air from the surrounding environment. The anticyclone would theoretically increase in strength as the surface height increased, as seen in Fig. 10. This assumes eddy mixing to be unchanged, which may explain the weakened anomaly in Iteration 10. The anomaly seen here differs from the more commonly considered atmospheric situation where advection of PV by the mean flow dominates horizontal mixing. A more thorough consideration of this simplified problem, including eddy mixing, mean flow, and planetary vorticity gradients,

is left as a subject for future work.

- More generally, the lateral boundary conditions for your model could be very different than what you specified due to climate system feedbacks as a result of reduced summer insolation so nesting a regional simulation in a GCM simulation for inception time might be the best next step in your modeling.

The revised manuscript emphasizes that the purpose of the study is to examine the role of regional feedbacks over the Baffin Island in leading to glacial inception. This indeed implies ignoring some large-scale feedbacks in order to focus on the role of those local feedbacks that are included. Furthermore, we are interested mostly in the very initial part of the inception, when one expects planetary-scale feedback to be less important in any case. The manuscript provides a detailed discussion of these issues throughout. To emphasize our focus on local feedbacks, we have renamed our manuscript to **"Role of Regional Feedbacks in Glacial Inception on Baffin Island: The Interaction of Ice Flow and Meteorology"**.

As an example of the new discussion added, the introduction now clearly states (Page 1, lines 18-24:

> The leading hypothesis for the cause of glacial inception is that orbital variations (Milankovitch, 1941) cause cooler summers leading to less melting, although previous studies found that additional amplifying feedbacks beyond the ice albedo feedback are necessary to explain inception. Possible amplifiers of orbital forcing may be divided into large-scale non-local feedbacks, including ocean temperature and planetary-scale circulation changes, and local feedbacks associated with greater height and extent of mountain glaciers (Lee and North, 1995; Oerlemans, 2002; Abe-Ouchi et al., 2007), regional circulation changes, clouds, and more. In this study, we seek to understand specifically the role of local ice-climate feedbacks in glacial inception using a regional climate model asynchronously coupled to a simple ice sheet model.

- Yet another rendition of the altered environment around 115k yr ago is Otieno et al. 2011: Atmospheric circulation anomalies due to 115k yr BP climate forcing are dominated by changes in the North Pacific Ocean. Clim. Dyn.

Thank you for bringing the Otieno et al. (2012) paper to our attention. We have found it very useful, and we believe it emphasizes that circulation over the Atlantic may not have changed much by the time of inception and cite in a few places in the revised manuscript, including in the discussion of ocean

temperature changes (page 21, lines 32-33),

[revised manuscript text omitted]

$$D_{i-\frac{1}{2},j}^n = A H_{i-\frac{1}{2},j}^{m+2} \left[ \left(\frac{H_{i,j}^* - H_{i-1,j}^*}{\Delta x_{i-1}}\right)^2 + \left(\frac{1}{2}\left(\frac{H_{i,j+1}^* - H_{i,j-1}^*}{\Delta y_{j-1} + \Delta y_j} + \frac{H_{i-1,j+1}^* - H_{i-1,j-1}^*}{\Delta y_{j-1} + \Delta y_j}\right)\right)^2 \right]^{(m-1)/2} \quad \text{(A5)}$$

$$D_{i,j+\frac{1}{2}}^n = A H_{i,j+\frac{1}{2}}^{m+2} \left[ \left(\frac{1}{2}\left(\frac{H_{i+1,j+1}^* - H_{i-1,j+1}^*}{\Delta x_{i-1} + \Delta x_i} + \frac{H_{i+1,j}^* - H_{i-1,j}^*}{\Delta x_{i-1} + \Delta x_i}\right)\right)^2 + \left(\frac{H_{i,j+1}^* - H_{i,j}^*}{\Delta y_j}\right)^2 \right]^{(m-1)/2} \quad \text{(A6)}$$

$$D_{i,j-\frac{1}{2}}^n = A H_{i,j-\frac{1}{2}}^{m+2} \left[ \left(\frac{1}{2}\left(\frac{H_{i+1,j+1}^* - H_{i-1,j+1}^*}{\Delta x_{i-1} + \Delta x_i} + \frac{H_{i+1,j}^* - H_{i-1,j}^*}{\Delta x_{i-1} + \Delta x_i}\right)\right)^2 + \left(\frac{H_{i,j}^* - H_{i,j-1}^*}{\Delta y_{j-1}}\right)^2 \right]^{(m-1)/2} \quad \text{(A7)}$$

Finally, our use of flux form requires Equation A11.

$$H_{i+\frac{1}{2},j}^{m+2} = \left(\frac{H_{i+1,j} + H_{i,j}}{2}\right)^{m+2} \quad \text{(A8)}$$

$$H_{i-\frac{1}{2},j}^{m+2} = \left(\frac{H_{i,j} + H_{i-1,j}}{2}\right)^{m+2} \quad \text{(A9)}$$

$$H_{i,j+\frac{1}{2}}^{m+2} = \left(\frac{H_{i,j+1} + H_{i,j}}{2}\right)^{m+2} \quad \text{(A10)}$$

$$H_{i,j-\frac{1}{2}}^{m+2} = \left(\frac{H_{i,j} + H_{i,j-1}}{2}\right)^{m+2} \quad \text{(
[revised manuscript text omitted]